# Learning Partial Graph Matching via Optimal Partial Transport

**Gathika Ratnayaka[1], James Nichols[2], & Qing Wang[1],**

[1] School of Computing, Australian National University, Australia

[2] Biological Data Science Institute, Australian National University, Australia

`{gathika.ratnayaka,james.nichols,qing.wang}@anu.edu.au`

## Abstract

Partial graph matching extends traditional graph matching by allowing some nodes to remain unmatched, enabling applications in more complex scenarios. However, this flexibility introduces additional complexity, as both the subset of nodes to match and the optimal mapping must be determined. While recent studies have explored deep learning techniques for partial graph matching, a significant limitation remains: the absence of an optimization objective that fully captures the problem's intrinsic nature while enabling efficient solutions. In this paper, we propose a novel optimization framework for partial graph matching, inspired by optimal partial transport. Our approach formulates an objective that enables partial assignments while incorporating matching biases, using weighted total variation as the divergence function to guarantee optimal partial assignments. Our method can achieve efficient, exact solutions within cubic worst case time complexity. Our contributions are threefold: (i) we introduce a novel optimization objective that balances matched and unmatched nodes; (ii) we establish a connection between partial graph matching and linear sum assignment problem, enabling efficient solutions; (iii) we propose a deep graph matching architecture with a novel partial matching loss, providing an end-to-end solution. The empirical evaluations on standard graph matching benchmarks demonstrate the efficacy of the proposed approach.

## 1 Introduction

Graph matching is a fundamental problem in network analysis, aiming to establish one-to-one correspondences between nodes in two graphs based on a defined objective. It has broad applications across various fields, including computer vision (Sun et al., 2020), bioinformatics (Zaslavskiy et al., 2009), and social network analysis (Zhang et al., 2019), where it is used to solve complex real-world problems. Traditional graph matching assumes a bijective mapping between the nodes of two graphs or a total injective mapping from the smaller graph to the larger one. However, these assumptions often restrict its applicability in more complex, real-world scenarios. *Partial graph matching*, a generalized version of the graph matching problem, addresses these limitations by allowing some nodes in both graphs to remain unmatched (Wang et al., 2023; Jiang et al., 2022b). It seeks an optimal partial assignment, an injective partial function between node sets, thereby expanding the practical utility of graph matching. This approach is particularly useful in applications where not all nodes have meaningful counterparts. For example, in image keypoint matching, not all keypoints in two given images have correspondences (Jiang et al., 2022b), and in biological networks, certain proteins may lack direct counterparts in other species (Zaslavskiy, 2010).

Solving partial graph matching is inherently more complex than traditional graph matching due to the additional challenge of determining both the subset of nodes to match and the optimal mapping itself. Graph matching is generally framed as a combinatorial optimization problem, where the choice of optimization objective is crucial. Early works formulated it as a Quadratic Assignment Problem (QAP), which is NP-hard (Koopmans & Beckmann, 1957; Lawler, 1963). Recent approaches (Wang et al., 2021; 2020a; Rolínek et al., 2020; Fey et al., 2020) use neural networks to learn node representations and derive a cross-graph node-to-node cost matrix. This allows graph matching to be framed as a linear sum assignment problem (Burkard et al., 2012; Yu et al., 2020), assuming a

bijective mapping between two node sets of equal size. Even if two graphs have an unequal number of nodes, the problem can still be framed as a linear sum assignment by assuming an injective mapping from the smaller graph to the larger one (Bonneel & Coeurjolly, 2019; Wang et al., 2019b). A key advantage of this approach is that it can be efficiently solved with cubic worst-case time complexity using the Hungarian algorithm (Kuhn, 1955). However, Partial graph matching does not assume a total mapping, adding an extra layer of complexity. It requires selecting the subset of nodes to be matched while determining the mapping, making the problem more challenging. Consequently, traditional linear assignment methods like linear sum or $k$ assignment cannot be directly applied.

Despite its challenging nature, recent studies have attempted to address the partial graph matching problem using deep learning techniques (Jiang et al., 2022b; Wang et al., 2023). One approach (Jiang et al., 2022b) frames the problem as an Integer Linear Programming (ILP) task with dummy nodes. However, ILPs rely on branch and bound algorithms, which lack polynomial worst-case time complexity, and the use of dummy nodes can hinder node representation learning (Wang et al., 2023). Another approach (Wang et al., 2023) estimates the number of matchings ($k$) between two graphs, solving it as a $k$-assignment problem (Burkard et al., 2012) using GreedyTopK algorithm (Wang et al., 2023). However, determining the optimal $k$ is difficult and requires separate neural modules, leading to multiple training stages. These limitations highlight the need for an efficient optimization objective that captures the inherent nature of partial graph matching.

**Present work** We address limitations in partial graph matching studies by proposing a novel optimization objective that finds an optimal partial mapping between two node sets while identifying which nodes should be matched. While optimal transport has been previously studied for graph matching (Xu et al., 2019; Saad-Eldin et al., 2021; Chen et al., 2020), its conservation of mass constraint, requiring all mass to be transported between distributions, limits its applicability to partial graph matching. In optimal partial transport, the mass conservation constraint is relaxed, allowing partial mass to be transported, with the optimal amount of mass determined during the optimization process (Séjourné et al., 2023; Bai et al., 2023). Although optimal partial transport and partial graph matching share similarities, i.e., both involve partial solutions, with the former identifying a plan where not all mass is transported and the latter finding a mapping where not all nodes are matched, they differ fundamentally. Partial graph matching requires the mapping to be an injective partial function (a partial assignment), while optimal partial transport does not. As a first step to bridging this fundamental difference, we reformulate partial graph matching as an optimal partial transport problem, where each node is assigned a unit mass and weighted total variation acts as the divergence function to dynamically balance matched and unmatched nodes. The optimization problem accomplishes two critical objectives: (i) it guarantees the existence of an optimal solution that induces a partial assignment, and (ii) it enables the incorporation of information related to relative significance of nodes to the matching process.

Based on these theoretical observations, we define a new optimization objective for partial graph matching. We demonstrate that the proposed optimization objective for partial graph matching can be solved by embedding it in a linear sum assignment problem. This enables solving the partial graph matching problem within a cubic worst case time complexity.

In summary, we make the following contributions in this work:

- We define a new optimization problem for partial graph matching, inspired by optimal partial transport. This formulation provides a robust optimization objective that carefully balances the selection of matched and unmatched nodes. Further, the formulation enables the incorporation of information related to the relative significance of each node to the matching process, which we term as matching bias.
- To solve the proposed optimization problem efficiently, we explore the underlying structure of its solution spaces. This reveals a notable embedding of the partial graph matching problem in a linear sum assignment problem. Furthermore, the solutions of the latter assignment problem can be mapped efficiently to solutions of the partial matching problem, which are themselves optimal. This allows us to solve the partial graph matching problem exactly within a cubic worst case time complexity.
- Building on the theoretical insights of our proposed optimization objective, we introduce a deep graph matching architecture that embeds feature and structural properties into the cross-graph node-to-node cost matrix and matching biases. This architecture provides an end-to-end

solution for the partial graph matching problem, incorporating a novel loss function called *partial matching loss*.

We conduct experiments to empirically validate our proposed approach on partial graph matching benchmarks. The results demonstrate the efficacy and efficiency of the proposed approach.

## 2 RELATED WORK

### 2.1 DEEP GRAPH MATCHING

Several graph matching methods leverage neural networks to learn matching-aware node embeddings (Jiang et al., 2022b; Rolínek et al., 2020; Wang et al., 2020a; Yu et al., 2020; Gao et al., 2021). These methods integrate cross-graph node affinity with feature and structural data to learn node embeddings. The learned embeddings are then used to derive cross-graph node-to-node affinities. Then, the soft correspondences between nodes are typically obtained by applying Sinkhorn normalization on the cost-graph node-to-node affinity matrix. Once the soft correspondence matrix is obtained, it is projected into the space of permutation like binary matrices to achieve one-to-one node correspondences using algorithms such as the Hungarian algorithm (Kuhn, 1955) and Stable Matching algorithm (Ratnayaka et al., 2023), assuming a total injective mapping from the smaller graph to the larger graph.

Optimal transport techniques have also been discussed to solve the graph matching problem (Xu et al., 2019), where each node in the source graph is matched to a node in the target graph. Moreover, optimal partial transport techniques have been applied to subgraph matching (Pan et al., 2024), where a preset fraction of mass (corresponding to a fixed number of nodes) from a smaller graph is matched to nodes in a larger graph. This approach constrains the matching process by specifying in advance how many nodes from the smaller graph must be matched. However, these approaches are not directly applicable to partial graph matching, which requires identifying corresponding nodes between two graphs without any prior assumptions about the number of nodes to be matched. Moreover, spectral methods have been successfully applied to both graph matching (Wang et al., 2019a) and also to compactly encode maps between graphs and subgraphs (Pegoraro et al., 2022). While these problems share similarities in attempting to find correspondences at node or graph level, they differ fundamentally from partial graph matching in their matching objectives and constraints.

So far, only a few deep learning approaches have addressed partial graph matching. Jiang et al. (2022b) introduces dummy nodes to bypass explicit match number estimation, framing the problem as an Integer Linear Programming (ILP) task. However, ILP with branch and bound suffers from high time complexity, and the use of dummy nodes can distort node representations by implying higher similarity with unmatched nodes. Wang et al. (2023) attempts to solve partial graph matching as a k-assignment problem. However, as the match count (k) between two graphs is not known in a partial graph matching problem, their approach to handle partial matchings include two steps, first estimating the match count (k) by using a separate neural module and solving an entropic optimal transport problem and then using the estimated k value to solve partial graph matching as a k-assignment problem. This two step approach increase computational complexity and error propagation. Another line of work (Nurlanov et al., 2023; Jiang et al., 2022a) to perform partial graph matching have explored universe graph representation learning, where a "universe" graph is constructed for each object class in keypoint-based image analysis. However, these methods require prior knowledge specific to each class, such as the exact number of distinct keypoints (nodes) in a class, making them difficult to generalize to graphs or images from previously unseen classes. In contrast, our work tackles a more general form of the partial graph matching problem, operating without class-specific assumptions and accommodating graphs from unseen classes with arbitrary structures.

### 2.2 OPTIMAL TRANSPORT AND THE ASSIGNMENT PROBLEM

Assignment problems, especially the linear sum assignment problem, can be modeled as an Optimal transport problem by considering assignment costs as transportation costs between two discrete distributions (Burkard et al., 2012). The Hungarian algorithm (Kuhn, 1955) which is a well-known methods for solving the linear sum assignment problem, operates with a worst-case time complexity of $O(n^3)$, and is widely used in many real world applications (Munkres, 1957; Yu et al., 2020).

In contrast to the linear sum assignment problem, partial graph matching can be considered an assignment problem where elements in both sets can remain unassigned. This characteristic makes conventional optimal transport methods unsuitable for solving partial assignment problems. While efforts have been made to adapt optimal partial transport for partial assignments (Bai et al., 2023; Bonneel & Coeurjolly, 2019), these methods have limitations. (Bai et al., 2023) propose an optimal partial transport formulation that can guarantee an optimal solution inducing a partial assignment between two sets, but their algorithm is restricted to one-dimensional data with convex cost metrics, whereas partial graph matching typically involves higher-dimensional data and requires more flexible cost functions. Moreover, their formulation does not account for matching bias of elements, which is crucial in partial graph matching, especially in data-driven approaches where certain elements should be prioritized for assignment. Efficient algorithms ($\leq (O(n^3))$) for solving partial assignments in higher-dimensional spaces with general cost functions have not yet been developed, leaving a gap in the literature.

## 3 BACKGROUND

**Notations**  We denote the set of source elements as $W_S$ and the set of target elements as $W_T$. We also use $\| \cdot \|_1$ to refer the $L^1$ norm, $\langle, \rangle_F$ to refer the Frobenius inner product, and $\mathbf{1}_n$ to denote a column vector of ones with $n$ elements. For any given matrix $\pi \in \mathbb{R}^{m \times n}$, $\pi_1$ and $\pi_2$ denotes marginals of $\pi$ where $\pi_1 = \pi \mathbf{1}_n$ and $\pi_2 = \pi^\intercal \mathbf{1}_m$.

**Optimal Transport (Balanced)**  Let $\mu \in \mathbb{R}^n_{\geq 0}$ and $\nu \in \mathbb{R}^m_{\geq 0}$ be two non-negative vectors representing distributions with equal total mass, such that $\|\mu\|_1 = \|\nu\|_1$. We are also given a *cost* matrix $C \in \mathbb{R}^{m \times n}$, where $C_{ij}$ denotes the cost of transporting a unit of mass from location $i$ in the source set $W_S$ to location $j$ in the target set $W_T$. The *(balanced) optimal transport problem* considers all possible transport plans $\Pi(\mu, \nu) = \{\pi \in \mathbb{R}^{m \times n}_{\geq 0} | \pi \mathbf{1}_n = \mu, \pi^\intercal \mathbf{1}_m = \nu\}$ and is defined as,

$$\mathrm{OT}(\mu, \nu) := \min_{\pi \in \Pi(\mu, \nu)} \langle \pi, C \rangle_F. \tag{1}$$

The marginal conditions $\pi \mathbf{1}_n = \mu$ and $\pi^\intercal \mathbf{1}_m = \nu$ impose the mass conservation constraint, ensuring that the total mass is transported from one distribution to the other.

**Optimal Partial Transport**  In the optimal partial transport problem, the mass conservation constraint is relaxed, allowing partial mass transportation between distributions. While some studies (Chapel et al., 2020; Figalli, 2010) address predefined amounts of partial mass, we consider the optimal partial transportation problem where the amount of partial mass being transported is determined by the optimization objective.

Let $\Pi_{\leq}(\mu, \nu)$ be the set of admissible *partial* transport plans defined as

$$\Pi_{\leq}(\mu, \nu) = \{\pi \in \mathbb{R}^{m \times n}_{\geq 0} | \pi \mathbf{1}_n \leq \mu, \pi^\intercal \mathbf{1}_m \leq \nu\}. \tag{2}$$

The *optimal partial transport problem* is usually defined as

$$\mathrm{OPT}_\rho(\mu, \nu) := \min_{\pi \in \Pi_{\leq}(\mu, \nu)} \langle \pi, C \rangle_F + \rho D(\pi_1 | \mu) + \rho D(\pi_2 | \nu). \tag{3}$$

Here, $\rho > 0$, which is also termed as the unbalancedness parameter, is used to control the tolerance for destroying or creating mass, and $D(\cdot | \cdot)$ represents a divergence between two discrete measures. The term $\langle \pi, C \rangle_F$ captures the cost of transporting the mass by $\pi$ while $\rho D(\pi_1 | \mu) + \rho D(\pi_2 | \nu)$ accounts for the mass that has not been transported.

In the optimal transport literature, Total Variation (TV) and Kullback-Leibler (KL) divergence are commonly used divergence functions (Séjourné et al., 2023). Using TV as the divergence function in optimal partial transport is known to identify zero entries in the optimal plan based on the cost matrix (Bai et al., 2023; Séjourné et al., 2023).

## 4 PARTIAL GRAPH MATCHING PROBLEM

We represent a graph as $\mathcal{G} = (V, E)$, where $V$ is the set of nodes and $E$ is the set of edges. Let $\mathcal{G}_S = (V_S, E_S)$ be the source graph and $\mathcal{G}_T = (V_T, E_T)$ be the target graph for matching, with $|V_S| = m$ and $|V_T| = n$. Without loss of generality, we can assume $m \leq n$.

In this work, we tackle the partial graph matching problem through the lens of optimal partial transport, which offers a flexible and efficient framework for dealing with graphs of different sizes and structures. To this end, we define an objective function that is well suited to our partial graph matching task. We first state this objective as a general optimal transport problem, then adapt it to our setting of graph matching.

We represent $V_S$ and $V_T$ as mass vectors $\mu \in \mathbb{R}^m$ and $\nu \in \mathbb{R}^m$ respectively. We also assume a cost matrix $C \in \mathbb{R}^{m \times n}$, where $C_{ij}$ indicates the cost of moving a unit mass from $i \in V_S$ to $j \in V_T$. We take inspiration from the use of the objective function defined in Eq. (3), and adapt it to our purposes. In particular, we use a weighted *total variation* (TV) divergence, which both ensures sparse optimal transport plans, and furthermore allows us to incorporate a *matching bias* to each node in the source and target graphs. The matching bias weights are given by two vectors, $\alpha \in \mathbb{R}_{\geq 0}^m$ for nodes in $V_S$ and $\beta \in \mathbb{R}_{\geq 0}^n$ for nodes in $V_T$, finally resulting in a objective function given by

$$\mathrm{TC}(\pi; C, \alpha, \beta) := \langle \pi, C \rangle_F + \rho \left( \langle \alpha, \mu - \pi_1 \rangle + \langle \beta, \nu - \pi_2 \rangle \right). \tag{4}$$

As we consider marginal constraints $\pi \mathbf{1}_n \leq \mu$ and $\pi^\intercal \mathbf{1}_m \leq \nu$, the weighted total variation divergences can be written as $\langle \alpha, \mu - \pi_1 \rangle$, and $\langle \beta, \nu - \pi_2 \rangle$. In the general transport setting our set of feasible partial plans would be $\Pi_{\leq}(\mu, \nu)$ as defined as in Eq. (2), and the optimal partial transport problem with weighted total variation as the divergence function then is

$$\mathrm{WOPT}_\rho(\mu, \nu) := \min_{\pi \in \Pi_{\leq}(\mu, \nu)} \mathrm{TC}(\pi; C, \alpha, \beta). \tag{5}$$

It should be noted that in our partial graph matching method discussed in Section 7, the cost matrix $C$, and matching bias values $\alpha$ and $\beta$ are learned from data. In our setting however, we require optimal partial *assignments* rather than plans, meaning that each node in one graph is matched to a node in the other graph or no node at all. We can define the set $\mathcal{M}$ of all $m \times n$ binary matrices representing possible partial assignments between $V_S$ and $V_T$ as follows

$$\mathcal{M} := \{\pi \in \{0, 1\}^{m \times n} \mid \forall i \in V_S, \sum_{j=1}^n \pi_{ij} \leq 1 \text{ and } \forall j \in V_T, \sum_{i=1}^m \pi_{ij} \leq 1\}.$$

For any $\pi \in \mathcal{M}$, $\pi_{ij} = 1$ if and only if node $i \in V_S$ is matched with node $j \in V_T$. Given the graphs $\mathcal{G}_S$ and $\mathcal{G}_T$, the partial graph matching problem seeks to find an optimal $\pi \in \mathcal{M}$ with respect to the objective given in Eq. (4), that is we are solving the

$$\mathrm{PGM}_\rho := \min_{\pi \in \mathcal{M}} \mathrm{TC}(\pi; C, \alpha, \beta). \tag{6}$$

Note that the partial assignments form a subset of the partial transport plans between $\mathbf{1}_m$ and $\mathbf{1}_n$, that is $\mathcal{M} \subset \Pi_{\leq}(\mathbf{1}_m, \mathbf{1}_n)$, and in fact we have that $\mathcal{M} = \Pi_{\leq}(\mathbf{1}_m, \mathbf{1}_n) \cap \{0, 1\}^{m \times n}$.

## 5    PARTIAL GRAPH MATCHING SOLUTIONS

In this section we demonstrate that partial assignments solutions of $\mathrm{PGM}_\rho$ (i.e. that sit in $\mathcal{M}$) do indeed occur in the solution set of $\mathrm{WOPT}_\rho(\mathbf{1}_m, \mathbf{1}_n)$, which we can consider to be a *relaxed* version of $\mathrm{PGM}_\rho$. We further note that in general solutions of $\mathrm{WOPT}_\rho(\mu, \nu)$ are not necessarily partial assignments as the marginal constraints might rule out solutions in $\mathcal{M}$.

First we consider the following theorem, which shows that our choice of weighted TV defined in Eq. (4) can distinguish between feasible and non-feasible assignments.

**Theorem 5.1** (Infeasible Assignments). *Let $\pi^\star \in \arg\min_{\Pi_{\leq}(\mu, \nu)} \mathrm{TC}(\pi; C, \alpha, \beta)$ be any optimal solution of Eq. (5). For any $1 \leq i \leq m$ and $1 \leq j \leq n$ we have that $C_{ij} > \rho(\alpha_i + \beta_j) \implies \pi_{ij}^\star = 0$.*

From this theorem, we define $C_{ij} > \rho(\alpha_i + \beta_j)$ as the *feasibility condition* for $i$ and $j$. If the cost $C_{ij}$ exceeds this threshold, no mass will be transported between $i$ and $j$. When considering graph matching, this means that no assignment will occur between $i \in V_S$ and $j \in V_T$.

Now we consider the unit-weighted case, that is when $\mu = \mathbf{1}_m$ and $\nu = \mathbf{1}_n$. The following theorem demonstrates that there is at least one solution of Eq. (5) that is a partial assignment between $V_S$ and $V_T$, i.e. is a solution of Eq. (6).

**Theorem 5.2** (Existence of Optimal Solution). *If $\mu = \mathbf{1}_m$, $\nu = \mathbf{1}_n$ there exists an optimal plan $\pi^\star \in \arg\min_{\Pi_\le(\mu,\nu)} \mathrm{TC}(\pi; C, \alpha, \beta)$ such that $\pi^\star \in \mathcal{M}$.*

For the remainder of the paper we consider $\mu = \mathbf{1}_m$, $\nu = \mathbf{1}_n$, and to distinguish our objective function in this case, we write $\widehat{TC}$ to denote the total cost in this unit-weight setting, that is

$$\widehat{TC}(\pi; C, \alpha, \beta) := \langle \pi, C \rangle_F + \rho \left( \langle \alpha, \mathbf{1}_m - \pi_1 \rangle + \langle \beta, \mathbf{1}_n - \pi_2 \rangle \right). \tag{7}$$

From Theorem 5.2, when $\widehat{TC}(\pi; C, \alpha, \beta)$ is the objective function of Eq. (5), there exists an optimal plan $\pi \in \mathcal{M}$ that ensures each node is either matched with at most one element or left unmatched, inducing a valid partial matching between $V_S$ and $V_T$. Moreover, based on Theorem 5.1, if $\alpha_p > \alpha_q$ for two source elements $p$ and $q$, then $p$ is more likely to satisfy the feasibility condition with each target element compared to $q$, giving $p$ a higher chance of being assigned to a target element.

## 6 SOLVING THE PARTIAL GRAPH MATCHING PROBLEM

Although Theorem 5.2 shows that there is always an optimal transport plan that induces a valid solution to the partial graph matching problem, it does not specify how to find such a plan. In this section, we address this challenge. Our main result demonstrates that it is possible to derive a linear sum assignment problem for which a closed-form mapping exists from any of its given solution to a solution of the partial graph matching problem defined in Eq. (6). The significance of this result is that it allows us to adapt the celebrate Hungarian algorithm to solve the partial graph matching problem with cubic worst-case time complexity algorithm.

Given two sets of elements with equal cardinality, the linear sum assignment problem aims to find a bijective assignment that minimizes the total cost of assignment (Burkard et al., 2012). In order to demonstrate a connection between the partial graph matching problem and the linear sum assignment problem, we first create a set $V_D$ by appending $(n - m)$ dummy elements to $V_S$, thus making $|V_D| = |V_T| = n$. Let $\alpha_* > 0$ be any value such that $\alpha_* > \max_{1 \le i \le m} \alpha_i$. We define a cost matrix $\overline{C} \in \mathbb{R}^{n \times n}$ s.t.,

$$\overline{C}_{ij} = \begin{cases} C_{ij}, & \text{if } 1 \le i \le m \text{ and } C_{ij} \le \rho(\alpha_i + \beta_j) \\ \rho(\alpha_i + \beta_j), & \text{if } 1 \le i \le m \text{ and } C_{ij} > \rho(\alpha_i + \beta_j) \\ \rho(\alpha_* + \beta_j), & \text{if } m < i \le n. \end{cases} \tag{8}$$

Let $\mathcal{P}_n = \{ \pi \in \{0,1\}^{n \times n} \,|\, \pi \mathbf{1}_n = \mathbf{1}_n, \pi^\mathsf{T} \mathbf{1}_n = \mathbf{1}_n \}$ denote the set of $n \times n$ permutation matrices. Then, we consider the following linear sum assignment problem,

$$\mathrm{LAP}(V_D, V_T, \overline{C}) := \min_{\overline{\pi} \in \mathcal{P}} \langle \overline{\pi}, \overline{C} \rangle_F. \tag{9}$$

We first establish an equivalence between the objective functions of the partial graph matching problem in Eq. (6) and the linear sum assignment problem in Eq. (9).

To this end we define a closed form mapping that will obtain valid solutions to the original partial graph matching problem Eq. (6) from solutions of the linear sum assignment problem Eq. (9). We write $h : \mathbb{R}^{n \times n} \to \mathbb{R}^{m \times n}$ to denote the mapping which for $1 \le i \le m$ and $1 \le j \le n$ is given by

$$[h(\overline{\pi})]_{ij} = \begin{cases} \overline{\pi}_{ij}, & \text{if } C_{ij} \le \rho(\alpha_i + \beta_j), \\ 0 & \text{otherwise.} \end{cases} \tag{10}$$

**Lemma 6.1** (Equivalence). *Given a cost matrix $C$ and the weights $\alpha$ and $\beta$, any permutation matrix $\overline{\pi} \in \mathcal{P}_n$ will have $h(\overline{\pi}) \in \mathcal{M}$ and $h(\overline{\pi})$ satisfies $\langle \overline{\pi}, \overline{C} \rangle_F = \widehat{TC}(h(\overline{\pi}); C, \alpha, \beta) + \rho(n - m)\alpha_*$.*

The following theorem effectively states that $h(\overline{\pi}^\star)$ will be a valid solution for our problem of interest.

**Theorem 6.2** (Optimal Solution). *If $\overline{\pi}^\star \in \arg\min_{\overline{\pi} \in \mathcal{P}_n} \langle \overline{\pi}, \overline{C} \rangle_F$, then $\pi^\star = h(\overline{\pi}^\star)$ is a solution of partial graph matching problem, that is $\pi^\star \in \arg\min_{\pi \in \mathcal{M}} \widehat{TC}(\pi; C, \alpha, \beta)$.*

The proofs of all the theorems and lemmas including Theorem 6.2 are provided in the appendix.

*Remark* 6.3. It is known that the linear sum assignment problem can be solved using the Hungarian algorithm, which has a cubic worst-case time complexity (Kuhn, 1955). Therefore, the worst case time complexity of solving the partial graph matching problem is $O(n^3)$.

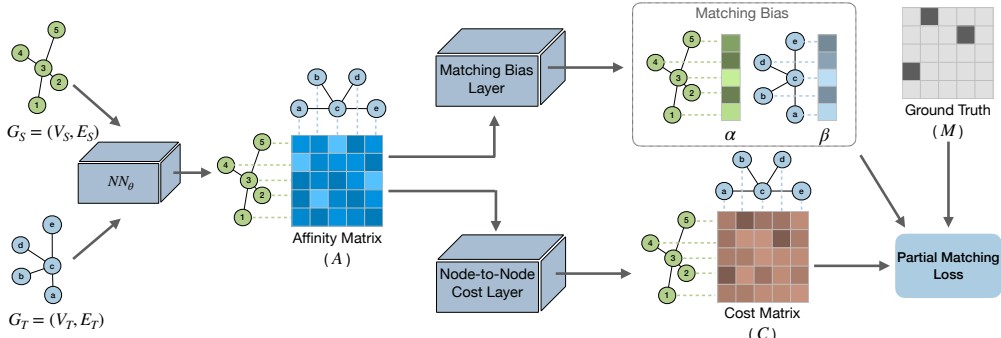

Figure 1: An overview of our end-to-end learning architecture for partial graph matching. Given two input graphs $G_S$ and $G_T$, the neural network $NN_\theta$ generates the cross-graph node-to-node affinity matrix $A$. The affinity matrix $A$ is then used to learn the matching bias values $\alpha$, $\beta$, and also the cost matrix $C$. The partial matching loss is computed by taking into account the ground truth, $C$, $\alpha$, and $\beta$.

# 7 PARTIAL GRAPH MATCHING ARCHITECTURE

In this section, we present an end-to-end learning architecture ( Fig. 1) for partial graph matching, building upon the theoretical properties discussed earlier. This architecture learns both the cost matrix $C$ and the matching biases of nodes ($\alpha$ and $\beta$). We also introduce a novel loss function, *partial matching loss*, designed to optimize performance in partial graph matching.

**Graph Affinity Encoding**  Graph affinity encoding uses a neural network $NN_\theta$ to transform geometric affinities between nodes into node embeddings, leveraging their features and structural information. These embeddings are then used to construct a matrix representing the cross-graph affinity between nodes in graphs $\mathcal{G}_S$ and $\mathcal{G}_T$. Specifically, given two input graphs $\mathcal{G}_S$ and $\mathcal{G}_T$, the neural network $NN_\theta : \mathbb{G} \times \mathbb{G} \to \mathbb{R}^{m \times n}$, parameterized by $\theta$, returns a cross-graph node-to-node affinity matrix $A \in \mathbb{R}^{m \times n}$. Essentially, the neural network $NN_\theta$ first applies an embedding function $f_{emb} : \mathbb{G} \to \mathbb{R}^{m \times d}$ to generate node embeddings $f_{emb}(\mathcal{G}_S) \in \mathbb{R}^{m \times d}$ and $f_{emb}(\mathcal{G}_T) \in \mathbb{R}^{n \times d}$ for the graphs $\mathcal{G}_S$ and $\mathcal{G}_T$, respectively. Then, the affinity function $f_{aff} : \mathbb{R}^{m \times d} \times \mathbb{R}^{n \times d} \to \mathbb{R}^{m \times n}$ combines these embeddings to compute a cross-graph node-to-node affinity matrix such that

$$NN_\theta(\mathcal{G}_S, \mathcal{G}_T) = f_{aff}\left(f_{emb}(\mathcal{G}_S), f_{emb}(\mathcal{G}_T)\right).$$

**Matching Biases and Cost**  The matching biases $\alpha$ and $\beta$ are calculated based on the cross-graph node-to-node affinity matrix $A$. Let $A^+ \in \mathbb{R}^{m \times n}_{\geq 0}$ be defined as $A^+_{ij} = \max\{A_{ij}, 0\}$. For each node $i \in V_S$, define $r_i = \max_{1 \leq j \leq n} A^+_{ij}$. Similarly, for each node $j \in V_T$, define $r_j = \max_{1 \leq i \leq m} A^+_{ij}$. The matching bias is then calculated as $\alpha_i = 2 \times (\sigma(w_{rs} \times r_i) - 0.5)$ for each $i \in V_S$ and $\beta_j = 2 \times (\sigma(w_{rs} \times r_j) - 0.5)$ for each $j \in V_T$, where $\sigma(\cdot)$ is the sigmoid function, and $w_{rs} \geq 0$ is a learnable parameter. A value of $\alpha_i$ or $\beta_j$ closer to 1 indicates a higher chance of node $i$ or $j$ being matched, while a value closer to 0 indicates a lower chance. Similarly, for $\beta_j$, a value closer to 1 suggests a higher chance of node $j$ being matched, and a value closer to 0 suggests a lower chance.

We compute the cost matrix $C$ between the nodes of $\mathcal{G}_S$ and $\mathcal{G}_T$ by applying Sinkhorn normalization (Sinkhorn, 1964) to the cross-graph node-to-node affinity matrix $A$. This normalization transforms $A$ into a doubly stochastic matrix $S \in [0, 1]^{m \times n}$ such that $S\mathbf{1}_n = \mathbf{1}_m$ and $S^\mathsf{T}\mathbf{1}_m \leq \mathbf{1}_n$ (Knight, 2008). The cost matrix $C \in [0, 1]^{m \times n}$ is then derived from $S$, with each element defined as $C_{ij} = 1 - S_{ij}$ for all nodes $i$ in $\mathcal{G}_S$ (where $1 \leq i \leq m$) and $j$ in $\mathcal{G}_T$ (where $1 \leq j \leq n$).

**Partial Matching Loss**  We have two primary learning objectives during training: (1) learning the matching cost matrix $C$, and (2) learning the matching biases $\alpha$ and $\beta$ of nodes in the source and target graphs. Below, we propose a new loss function that integrates these two learning objectives.

Let $\mathbb{I}(\cdot)$ denote the indicator function. Using the feasibility condition from Theorem 5.1, we define the matching attention matrix $Z$ as:

$$Z_{ij} = \mathbb{I}\left(M_{ij} = 1 \text{ or } C_{ij} \leq \rho(\alpha_i + \beta_j)\right).$$

Based on $Z$ and the ground truth matching matrix $M \in \mathcal{M}$, we propose the loss term $L_{cost}$ to guide the learning of the matching cost matrix $C$:

$$L_{cost} = -\sum_{i,j} Z_{ij} \left[ M_{ij} \log(1 - C_{ij}) + (1 - M_{ij}) \log(C_{ij}) \right].$$

A node pair is included in the loss $L_{cost}$ under two conditions: (1) The ground truth indicates the pair should be matched ($M_{ij} = 1$), or (2) The pair should not be matched ($M_{ij} = 0$) but the matching cost is below the feasibility condition ($C_{ij} \leq \rho(\alpha_i + \beta_j)$). According to Theorem 5.1, pairs exceeding the threshold are infeasible and thus are not penalized in the loss, as they cannot produce false positives.

To guide the learning of the matching biases $\alpha$ and $\beta$, we propose the loss term $L_{bias}$:

$$L_{bias} = \sum_{i=1}^{m} (M\mathbf{1}_n)_i \left[ 1 - \alpha_i \right]^2 + \sum_{j=1}^{n} (M^\intercal \mathbf{1}_m)_j \left[ 1 - \beta_j \right]^2.$$

The goal of $L_{bias}$ is to increase the matching bias of a node that should be matched. The partial matching loss is defined as $L = L_{cost} + \lambda L_{bias}$, where $\lambda \in (0, 1]$ is the regularization parameter.

## 8 EXPERIMENTS

We evaluate the performance and robustness of our proposed approach through several experiments: (1) We evaluate the efficacy of our approach through experiments on image keypoint matching datasets and Protein-Protein Interaction (PPI) networks under varying noise levels; (2) We analyze the effects of the matching biases $\alpha$ and $\beta$ on overall performance by considering two model variants: OPGM, with fixed equal matching biases ($\alpha = \mathbf{1}_m, \beta = \mathbf{1}_n$), and OPGM-rs, with learnable matching biases. Note that when training OPGM, $L = L_{cost}$, as we do not need to learn matching biases. (3) We conduct a sensitivity analysis on the unbalancedness parameter $\rho$ and regularization parameter $\lambda$ to understand their impact on partial graph matching performance; (4) We analyze the runtime efficiency of our approach compared to other baselines. Our code is available at GitHub: https://github.com/Gathika94/OPGMrs.git.

### 8.1 EXPERIMENTAL SETUP

**Image Keypoint Matching** In this task, we focus on image keypoint matching, which aims to find corresponding annotated keypoints between two given images. We use three image keypoint matching datasets that inherently contain outliers: 1) the *Pascal VOC Keypoint with Berkeley annotations* (Everingham et al., 2010), which includes keypoint-annotated images from 20 classes; 2) *SPair-71k* (Min et al., 2019), featuring 70,958 high-quality image pairs from Pascal VOC 2012 and Pascal 3D+, covering 18 classes; and 3) *IMC PT SparseGM* (Wang et al., 2023), a recently proposed dataset specifically for partial graph matching.

*Experimental Setting.* We follow the same experimental setting in (Wang et al., 2023). A pre-trained VGG16 model (Simonyan & Zisserman, 2014) is used to extract visual features of annotated image keypoints. Graphs are created using the extracted keypoint features following the same protocol as in (Wang et al., 2023). We use the neural network proposed in GCAN (Jiang et al., 2022b) as NN$_\theta$ to learn node embeddings and compute the cross-graph node-to-node affinity matrix.

Consistent with (Wang et al., 2023), we report the matching F1-score as the evaluation metric. When reporting the results, we run 5 random starts and report the 95% confidence interval as the error bars, similarly to (Wang et al., 2023). We considered the following baselines: NGM-v2 (Wang et al., 2020b), GCAN (Jiang et al., 2022b), AFAT-U (Jiang et al., 2022b), and AFAT-I (Wang et al., 2023). We consider the implementation and results reported for baselines in (Wang et al., 2023)[1] for evaluations. In Table 1, and Table 2(left), GM Network denotes the neural network architecture that each of the given baselines used to learn node embeddings and to obtain the cross-graph node-to-node affinity matrix ($A$). PMH indicates the technique that is used to distinguish matching vs non-matching nodes and to obtain the final partial matching.

---

[1]https://github.com/Thinklab-SJTU/ThinkMatch

| GM Network | PMH | aero | bike | bird | boat | bottle | bus | car | cat | chair | cow | dog | horse | mbike | person | plant | sheep | train | tv | mean |
|---|---|---|---|---|---|---|---|---|---|---|---|---|---|---|---|---|---|---|---|---|
| NGM-v2 | dummy | 47.7 | 41.6 | 62.1 | 30.3 | 59.0 | 49.7 | 27.4 | 68.3 | 33.9 | 62.4 | 57.3 | 46.7 | 46.4 | 42.7 | 78.7 | 43.5 | 80.5 | 89.5 | 53.8 ±0.4 |
| NGM-v2 | AFAT-U | 50.3 | 43.5 | 63.8 | **32.4** | 59.0 | 60.1 | **39.7** | **68.6** | 36.1 | 63.6 | 56.5 | 46.3 | **51.4** | 43.3 | 77.0 | **51.2** | 81.1 | 89.4 | 56.3 ±0.4 |
| NGM-v2 | AFAT-I | 50.4 | 43.6 | 63.9 | 32.1 | 61.2 | 58.5 | 38.0 | 68.4 | 35.7 | 62.7 | 56.4 | 47.7 | **51.9** | **44.3** | 78.5 | **50.7** | 79.2 | 91.2 | 56.4 ±0.4 |
| GCAN | ILP | 49.0 | 41.3 | 64.0 | 30.3 | 57.3 | 55.0 | 37.4 | 64.8 | 36.6 | 63.0 | 58.0 | 44.4 | 46.4 | 42.6 | 68.4 | 42.3 | 83.2 | **91.9** | 54.2 ±0.3 |
| GCAN | AFAT-U | 46.7 | 43.3 | 65.8 | **33.3** | **61.5** | 54.9 | 35.2 | 68.4 | 37.7 | 59.9 | 56.0 | 47.6 | 47.2 | 43.5 | **80.3** | 47.7 | **83.8** | 89.0 | 55.7±0.4 |
| GCAN | AFAT-I | 46.8 | **44.3** | 65.9 | **32.4** | **61.5** | 53.8 | 33.7 | 68.4 | **38.1** | 60.1 | 56.3 | **47.9** | 48.3 | 43.8 | **81.2** | 48.4 | 82.9 | 88.0 | 55.7 ±0.4 |
| GCAN | OPGM | **51.9** | **43.8** | **66.6** | 28.9 | 60.9 | **60.6** | 37.8 | 67.8 | 37.7 | **64.3** | **58.9** | 47.6 | 47.8 | 43.1 | 77.3 | 49.5 | 82.1 | 90.9 | **56.5 ±0.4** |
| GCAN | OPGM-rs | **53.0** | 43.5 | **66.7** | 32.1 | **61.7** | 61.4 | 40.7 | **68.8** | 38.5 | **65.8** | 59.5 | 51.1 | 47.2 | **46.2** | 78.6 | 48.7 | **83.3** | **92.3** | **57.7 ±0.2** |

Table 1: Performance (matching F1-score) on the dataset SPair-71K. The best results are colored in **black** and the second best are in **blue**.

| Dataset Name | | IMCPT 50 | | | | IMCPT 100 | | | |
|---|---|---|---|---|---|---|---|---|---|
| GM Network | PMH | reichstag | sacre | st peters | mean | reichstag | sacre | st peters | mean |
| NGM-v2 | dummy | 88.5 | 56.1 | 63.0 | 69.2±0.5 | 80.0 | 57.0 | 71.3 | 69.5±0.3 |
| NGM-v2 | AFAT-U | 90.5 | 58.7 | 66.9 | 72.0±0.3 | 81.7 | 57.0 | 72.2 | 70.3±0.2 |
| NGM-v2 | AFAT-I | **92.3** | 58.7 | 66.7 | **72.8±0.4** | 82.0 | 57.0 | 71.4 | 70.1±0.3 |
| GCAN | ILP | 87.2 | 55.1 | 63.0 | 68.4±0.5 | 80.4 | 55.7 | 72.8 | 69.6±0.4 |
| GCAN | AFAT-U | 86.9 | **59.4** | 67.1 | 71.1±0.4 | **82.6** | 58.2 | 73.8 | 71.5±0.2 |
| GCAN | AFAT-I | 91.0 | **60.3** | **67.3** | **72.9±0.6** | **82.7** | 57.8 | 72.4 | 70.9±0.4 |
| GCAN | OPGM | 91.2 | 57.0 | **68.1** | 72.1±0.2 | 81.7 | **59.1** | 76.1 | **72.3±0.4** |
| GCAN | OPGM-rs | **91.9** | 59.2 | 67.0 | 72.7±0.1 | 82.3 | **60.5** | 75.7 | **72.8±0.1** |

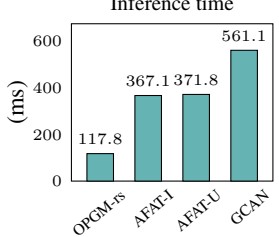

Table 2: (left) Performance (matching F1-score) on the datasets IMCPT 50 and IMCPT 100, where the best results are colored in **black** and the second best are in **blue**; (right) Inference time of our model OPGM-rs against the state-of-the-art methods AFAT-I, AFAT-U, and GCAN on the dataset IMCPT 100.

**PPI Network Matching**  In this task, we focus on Protein Protein Interaction network matching under varying noise levels. PPI network matching dataset is a standard graph matching benchmark that can be used to evaluate performance of a graph matching model under various noise levels (Liu et al., 2021; Ratnayaka et al., 2023). It is a protein protein interaction (PPI) network of yeasts, consisting of 1004 proteins and 4920 high-confidence interactions among those proteins. The PPI network matching problem is to match this network with its noisy versions, which contain 5%, 10%, 15%,20%,25% additional interactions (low-confidence interactions), respectively.

*Experimental Setting.*  We adopt the experimental settings of the baseline methods SIGMA (Liu et al., 2021) and StableGM (Ratnayaka et al., 2023) and use node correctness (the percentage of nodes that have the same matching as the ground truth) as the evaluation metric, similarly to (Liu et al., 2021; Ratnayaka et al., 2023). The neural network architecture $\text{NN}_\theta$ follows the design used in these baselines, where each node's input feature is derived from its degree. A 5-layer Graph Isomorphism Network (GIN) (Xu et al., 2018) is employed. Node-to-node affinities across graphs are computed using the cosine similarities of node embeddings. Note that PPI network matching focuses on finding a bijective mapping between two graphs with an equal number of nodes. As shown in Theorem 5.1, our method achieves bijective mapping for sufficiently large $\rho$. Hence, we set $\rho = 10^{11}$ for this task.

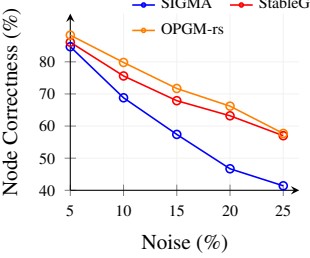
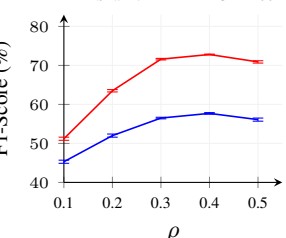
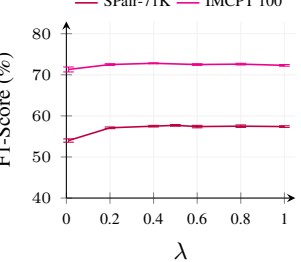

Figure 2: (left) Performance (node correctness) of our model OPGM-rs against SIGMA and StableGM on PPI Network Matching with varying noise levels; (middle) Sensitivity analysis of our model OPGM-rs on the unbalancedness parameter $\rho$; (right) Sensitivity analysis of our model OPGM-rs on the regularization parameter $\lambda$. Performance in sensitivity analysis is measured by matching F1-score.

## 8.2 RESULTS AND DISCUSSION

**Ext-1 Architecture efficacy** Table 1 presents the results for the Spair-71K dataset (Table 1). Both OPGM and OPGM-rs outperform other baselines in terms of mean F1-score and achieve superior results in 11 out of 18 classes. The Spair-71K dataset provides several advantages over the Pascal VOC Keypoint dataset, including higher image quality, richer keypoint annotations, and the removal of ambiguous annotations (Rolínek et al., 2020). Evaluations on the Pascal VOC Keypoint dataset are discussed in the appendix, where our models perform slightly worse than some baselines, likely due to the poor and ambiguous annotations in Pascal VOC (Rolínek et al., 2020). These results highlight the effectiveness of our approach for visual graph matching, particularly on high-quality datasets like Spair-71K.

Table 2 (left) presents the results for the IMCPT 50 and IMCPT 100 datasets. On the IMCPT 100 dataset, our models achieve higher mean F1-scores than the baselines and outperform them in 2 out of 3 classes. For the IMCPT 50 dataset, our method performs comparably to the best baselines. Notably, the IMCPT 100 dataset, with the largest number of nodes among visual graph matching datasets (Wang et al., 2023), showing the efficacy of our approach for partial graph matching.

As shown in Fig. 2 (left), OPGM-rs consistently outperforms other baselines across all noise levels for PPI network matching (see the appendix for numerical results). However, at higher noise levels, the performance gap between OPGM-rs and StableGM narrows.

**Ext-2 Impact of matching biases** From the results given in Tables 1 and 2 and Fig. 2 (left), it is clear that OPGM-rs consistently outperforms OPGM in most cases, highlighting the importance of learning the matching biases of nodes in partial graph matching.

**Ext-3 Sensitivity analysis** We evaluated the impact of the unbalancedness parameter $\rho$ and the regularization parameter $\lambda$ on partial graph matching performance using Spair-71K and IMCPT 100 datasets. For $\rho$, we varied its values while keeping other hyperparameters fixed. As shown in Fig. 2 (middle), mean F1-scores highlight $\rho$'s critical role: smaller values restrict valid matches, while larger values may allow incorrect matches, both reducing performance. For $\lambda$, starting from $0.01$, we tested $\lambda \in \{0.2, 0.4, 0.6, 0.8, 1\}$ with fixed hyperparameters. As shown in Fig. 2 (right), the mean F1-scores show slight variation, indicating that the impact of $\lambda$ is not significant.

**Ext-4 Efficiency analysis** We evaluated the average runtime for processing a pair of graphs (i.e., keypoint-annotated images) during the inference phase for OPGM-rs, GCAN, AFAT-I, and AFAT-U using the IMCPT 100 dataset, the largest image keypoint matching dataset (Wang et al., 2023). Matching was performed on 3,000 keypoint-annotated pairs. As shown in Table 2 (right), OPGM-rs demonstrates significantly lower inference time compared to the other models. It is important to note that all models use the same neural network architecture from (Jiang et al., 2022b) (corresponding to $\text{NN}_\theta$) to compute the cross-graph node-to-node affinity matrix $A$ and the Sinkhorn algorithm to derive the doubly stochastic affinity matrix $S$, differing only in their partial matching techniques.

## 9 CONCLUSION AND LIMITATIONS

In this work, we proposed a new problem formulation for partial graph matching based on optimal partial transport. Our approach can dynamically determine which nodes would be matched or left unmatched during the optimization process. We demonstrate how our problem formulation enabled solving the partial graph matching problem within cubic worst case time complexity. We then showed how our proposed solution for partial graph matching can be effectively integrated to a learning setting. Evaluations on various partial graph matching benchmarks demonstrate that our method outperform the baselines in most of the benchmarks. Beyond these specific tasks discussed in this work, the optimization problem we proposed and the solution mechanism we developed can be adapted to any application requiring optimal partial assignment between two sets.

While our proposed method demonstrated strong performance on datasets with reliable annotations, its effectiveness diminished when faced with unreliable or ambiguous annotations, or higher noise levels in the data. Enhancing the robustness of our method under these challenging conditions is an important direction for future work.

ACKNOWLEDGEMENTS

This research was supported partially by the Australian Government through the Australian Research Council's Discovery Projects funding scheme (project DP210102273). We also would like to thank anonymous reviewers for their comments, which helped improve the quality of the paper.

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

## A APPENDIX

### A.1 EQUIVALENCE BETWEEN OPTIMAL TRANSPORT AND OPTIMAL PARTIAL TRANSPORT

Inspired by (Bai et al., 2023; Caffarelli & McCann, 2010), we show that the optimal partial transport problem proposed in Eq. (5) has an equivalance with an optimal (balanced) transport problem. In the following we will write $\|\mu\|_1 = \sum_{i=1}^{m} |\mu_i|$ or $\|\pi\|_1 = \sum_{i,j=1}^{m,n} |\pi_{ij}|$ to denote the 1-norm or sum of elements of probability vectors and matrices noting that all $\mu_i, \pi_{ij} > 0$.

Let $K$ be a constant satisfying $K \geq \|\mu\|_1 + \|\nu\|_1$, and we define extended vectors $\hat{\nu}, \hat{\mu} \in \mathbb{R}^{m+n}$ as

$$\hat{\mu}_i = \begin{cases} \mu_i & \text{if } i \leq m, \\ \frac{1}{n}(K - \|\mu\|_1) & \text{if } m < i \leq m+n \end{cases} \quad \text{and} \quad \hat{\nu}_j = \begin{cases} \nu_j & \text{if } j \leq n, \\ \frac{1}{m}(K - \|\nu\|_1) & \text{if } n < j \leq m+n \end{cases}$$

and note that $\|\hat{\mu}\|_1 = \|\hat{\nu}\|_1 = K$. We define $\widehat{C} \in \mathbb{R}^{(m+n)\times(m+n)}$ such that

$$\widehat{C}_{ij} = \begin{cases} C_{ij} - \rho(\alpha_i + \beta_j) & \text{if } i \leq m \text{ and } j \leq n, \\ 0 & \text{if otherwise.} \end{cases} \tag{11}$$

We consider the following balanced optimal transport problem between the extended vectors $\hat{\mu}$ to $\hat{\nu}$ that optimizes over the set of admissible couplings $\Pi(\hat{\mu}, \hat{\nu}) = \{\pi \in \mathbb{R}_{\geq 0}^{(m+n)\times(m+n)} | \pi \mathbf{1}_{m+n} = \hat{\mu}, \pi^{\mathsf{T}} \mathbf{1}_{m+n} = \hat{\nu}\}$.

$$\text{OT}(\hat{\mu}, \hat{\nu}) = \min_{\pi \in \Pi(\hat{\mu}, \hat{\nu})} \langle \pi, \widehat{C} \rangle. \tag{12}$$

We claim that there exists an equivalence between this OT problem and Eq. (5).

**Proposition A.1** (Equivalent Cost of Couplings). *For any $\hat{\pi}, \hat{\pi}' \in \Pi(\hat{\mu}, \hat{\nu})$ with $\hat{\pi}[1:m][1:n] = \hat{\pi}'[1:m][1:n] = \pi$*

$$\langle \widehat{C}, \hat{\pi} \rangle_F = \langle \widehat{C}, \hat{\pi}' \rangle_F$$

*and*

$$\text{TC}(\pi; C, \alpha, \beta) = \langle \widehat{C}, \hat{\pi} \rangle_F + \rho(\|\alpha\|_1 + \|\beta\|_1).$$

*Proof.* Based on the definition of $\widehat{C}$, we can derive that for any $\pi \in \Pi(\hat{\mu}, \hat{\nu})$,

$$\langle \widehat{C}, \pi \rangle_F = \sum_{i=1}^{m} \sum_{j=1}^{n} \widehat{C}_{ij} \pi_{ij}$$

Together with the above result and the fact that $\hat{\pi}[1:m][1:n] = \hat{\pi}'[1:m][1:n] = \pi$, we can derive

$$\langle \widehat{C}, \hat{\pi} \rangle_F = \sum_{i=1}^{m} \sum_{j=1}^{n} \widehat{C}_{ij} \hat{\pi}_{ij} = \sum_{i=1}^{m} \sum_{j=1}^{n} \widehat{C}_{ij} \pi_{ij} = \sum_{i=1}^{m} \sum_{j=1}^{n} \widehat{C}_{ij} \hat{\pi}'_{ij} = \langle \widehat{C}, \hat{\pi}' \rangle_F \tag{13}$$

Thus, $\langle \widehat{C}, \hat{\pi} \rangle_F = \langle \widehat{C}, \hat{\pi}' \rangle_F$.

From Eq. (7), we know that,

$$\begin{aligned} \text{TC}(\pi; C, \alpha, \beta) &= \langle \pi, C \rangle_F + \rho\left( \langle \alpha, \mathbf{1}_n - \pi_1 \rangle + \langle \beta, \mathbf{1}_m - \pi_2 \rangle \right) \\ &= \sum_{i=1}^{m} \sum_{j=1}^{n} (C_{ij} - \rho(\alpha_i + \beta_j)) \pi_{ij} + \rho(\|\alpha\|_1 + \|\beta\|_1) \\ &= \langle \widehat{C}, \hat{\pi} \rangle_F + \rho(\|\alpha\|_1 + \|\beta\|_1) \end{aligned}$$

where in the last step we used Eq. (13). Thus, the proof is complete. $\square$

**Proposition A.2** (Equivalence of Minimizers). *For any $\hat{\pi} \in \Pi(\hat{\mu}, \hat{\nu})$ that minimizes equation 12 then the top left corner $\pi = \hat{\pi}[1:m][1:n]$ satisfies $\pi \in \Pi_{\leq}(\mu, \nu)$, and $\pi$ furthermore minimizes equation 5.*

*Proof.* For any $\hat{\pi} \in \Pi(\hat{\mu}, \hat{\nu})$, let $\hat{\pi}[1:m][1:n] = \pi$.

From the definitions of $\hat{\mu}$ and $\hat{\nu}$, we know that $\hat{\mu}[1:m] = \mu$ and $\hat{\nu}[1:n] = \nu$. Therefore, $\pi_1 \leq \mu$ and $\pi_2 \leq \nu$. Thus, $\pi \in \Pi_{\leq}(\mu, \nu)$.

From Proposition A.1, we know that,

$$\text{TC}(\pi; C, \alpha, \beta) = \langle \widehat{C}, \hat{\pi} \rangle_F + \rho(\|\alpha\|_1 + \|\beta\|_1).$$

We also know that for any $\hat{\pi} \in \Pi(\hat{\mu}, \hat{\nu})$, $\exists \pi \in \Pi_{\leq}(\mu, \nu)$ s.t. $\pi = \hat{\pi}[1:m][1:n]$. Therefore, $\text{TC}(\pi; C, \alpha, \beta)$ is minimized when $\langle \widehat{C}, \hat{\pi} \rangle_F$ is minimized. Consequently, for any $\hat{\pi} \in \Pi(\hat{\mu}, \hat{\nu})$ that minimizes Eq. (12) then the top left corner $\pi = \hat{\pi}[1:m][1:n]$ satisfies $\pi \in \Pi_{\leq}(\mu, \nu)$, and $\pi$ furthermore minimizes Eq. (5).

$\square$

## A.2 PROOFS

**Theorem 5.1** (Infeasible Assignments). *Let $\pi^{\star} \in \arg\min_{\Pi_{\leq}(\mu, \nu)} \text{TC}(\pi; C, \alpha, \beta)$ be any optimal solution of Eq. (5). For any $1 \leq i \leq m$ and $1 \leq j \leq n$ we have that $C_{ij} > \rho(\alpha_i + \beta_j) \implies \pi_{ij}^{\star} = 0$.*

*Proof.* When total variation is assumed as the divergence function $D(.|.)$, Eq. (4) can be written as

$$\text{TC}(\pi; C, \alpha, \beta) := \langle \pi, C \rangle_F + \rho\left(\langle \alpha, \mu - \pi_1 \rangle + \langle \beta, \nu - \pi_2 \rangle\right).$$

For any given source sample $p \in V_S$ and any given target sample $q \in V_T$, we can write $\text{TC}(\pi; C, \alpha, \beta)$ by separating the terms related to masses of $p \in V_S$ and $q \in V_T$ (See Eq. (7) as well).

$$
\begin{aligned}
\text{TC}(\pi; C, \alpha, \beta) = & \sum_{\substack{1 \leq i \leq m \\ i \neq p}} \sum_{\substack{1 \leq j \leq n \\ i \neq q}} \pi_{ij} C_{ij} + \sum_{\substack{1 \leq j \leq n \\ i \neq q}} \pi_{pj} C_{pj} + \sum_{\substack{1 \leq i \leq m \\ i \neq p}} \pi_{iq} C_{iq} + \pi_{pq} C_{pq} \\
& + \rho\left( \sum_{\substack{1 \leq i \leq m \\ i \neq p}} \alpha_i(\mu_i - (\pi_1)_i) + \alpha_p\left( \mu_p - \sum_{\substack{1 \leq j \leq n \\ i \neq q}} \pi_{pj} - \pi_{pq} \right) \right) \\
& + \rho\left( \sum_{\substack{1 \leq j \leq n \\ i \neq q}} \beta_j(\nu_j - (\pi_2)_j) + \beta_q\left( \nu_q - \sum_{\substack{1 \leq i \leq m \\ i \neq p}} \pi_{iq} - \pi_{pq} \right) \right)
\end{aligned}
$$

We consider two plans $\pi, \pi' \in \Pi_{\leq}(\mu, \nu)$ such that $\pi_{pq} \neq \pi'_{pq}$, but otherwise $\pi_{ij} = \pi'_{ij}$ for all other $1 \leq i \leq m$, $1 \leq j \leq n$. Note that this is possible as they are partial transport plans that do not have strict marginal equality constraints. In this case we find that

$$\text{TC}(\pi; C, \alpha, \beta) - \text{TC}(\alpha, \beta, \pi', C) = (\pi_{pq} - \pi'_{pq})(C_{pq} - \rho(\alpha_p + \beta_q))$$

We note that this implies strict monotonicity of the total transport cost if $C_{pq} - \rho(\alpha_p + \beta_q) > 0$, i.e. that $\text{TC}(\pi; C, \alpha, \beta) > \text{TC}(\alpha, \beta, \pi', C)$ if $\pi_{pq} > \pi'_{pq}$.

Therefore, if $C_{pq} - \rho(\alpha_p + \beta_q) > 0$ then the optimal strategy is always going to be to set $\pi_{pq} = 0$, ensuring the lowest total cost possible. In other words, if for any sample $p \in V_S$ and $q \in V_T$, $C_{pq} > \rho(\alpha_p + \beta_q)$, it is always less costly to transport no mass from $p$ to $q$.

Therefore, if $\pi^* \in \Pi_{\leq}(\mu, \nu)$ be any optimal solution of Eq. (5). Then, $\forall i \in V_S, \forall j \in V_T$, if $C_{ij} > \rho(\alpha_i + \beta_j)$ then $\pi_{ij}^{\star} = 0$.

$\square$

**Theorem 5.2** (Existence of Optimal Solution). *If $\mu = \mathbf{1}_m$, $\nu = \mathbf{1}_n$ there exists an optimal plan $\pi^\star \in \arg\min_{\Pi_{\leq}(\mu,\nu)} \mathrm{TC}(\pi; C, \alpha, \beta)$ such that $\pi^\star \in \mathcal{M}$.*

*Proof.* Let us define extended vectors $\hat{\nu} = \mathbf{1}_{m+n}, \hat{\mu} = \mathbf{1}_{m+n}$. Once again we make use of the matrix $\widehat{C}$ as defined in Eq. (11).

Next, we consider the following optimal transport problem that optimizes over the set of admissible couplings $\Pi(\hat{\mu}, \hat{\nu}) = \{\pi \in \mathbb{R}_{\geq 0}^{(m+n)\times(m+n)} | \pi \mathbf{1}_{m+n} = \hat{\mu} = \mathbf{1}_{m+n}, \pi^{\mathsf{T}} \mathbf{1}_{m+n} = \hat{\nu} = \mathbf{1}_{m+n}\}$.

$$\mathrm{OT}(\hat{\mu}, \hat{\nu}) = \min_{\pi \in \Pi(\hat{\mu}, \hat{\nu})} \langle \pi, \widehat{C} \rangle. \tag{14}$$

From the results of Appendix A.1, we can derive that the optimization problem $\mathrm{OT}(\hat{\mu}, \hat{\nu})$ given in Eq. (14) has a clear equivalence to the optimization problem $\mathrm{OPT}_\rho(\mu, \nu)$ given in Eq. (5) when $K = m + n$. From Appendix A.1, we also know that the optimal plan for Eq. (5) can be obtained by restricting an optimal plan $\hat{\pi} \in \arg\min_{\pi \in \Pi(\hat{\mu}, \hat{\nu})} \langle \pi, \widehat{C} \rangle$ to the set $[1:m] \times [1:n]$.

It should be noted that when $\hat{\nu} = \mathbf{1}_{m+n}$, and $\hat{\mu} = \mathbf{1}_{m+n}$, the set $\Pi(\hat{\mu}, \hat{\nu})$ corresponds to the set of doubly stochastic matrices. By the Birkhoff-von-Neumann theorem, it is known that the set of doubly stochastic matrices forms a convex polytope and its extremal points are permutation matrices. From the Linear Programming theory, it is known that at least one optimal solution to a linear program over a polytope (here, it is the set of doubly stochastic matrices) lies at an extreme point of the polytope. Thus, there always exist an optimal plan $\hat{\pi} \in \Pi(\hat{\mu}, \hat{\nu})$ which is a permutation matrix. Therefore, the optimal plan $\pi^\star \in \arg\min_{\Pi_{\leq}(\mu,\nu)} \mathrm{TC}(\pi; C, \alpha, \beta)$ that is obtained by restricting such $\hat{\pi} \in \arg\min_{\pi \in \Pi(\hat{\mu}, \hat{\nu})} \langle \pi, \widehat{C} \rangle$ (which is a permutation matrix) to its restriction to $[1:m] \times [1:n]$ (i.e., $\pi^* = \hat{\pi}[1:m][1:n]$) will be a binary matrix with atmost one 1 per row or column. Therefore, we can always find an optimal plan $\pi^\star \in \arg\min_{\Pi_{\leq}(\mu,\nu)} \mathrm{TC}(\pi; C, \alpha, \beta)$ s.t. $\pi^\star \in \mathcal{M}$. $\square$

**Lemma 6.1** (Equivalence). *Given a cost matrix $C$ and the weights $\alpha$ and $\beta$, any permutation matrix $\overline{\pi} \in \mathcal{P}_n$ will have $h(\overline{\pi}) \in \mathcal{M}$ and $h(\overline{\pi})$ satisfies $\langle \overline{\pi}, \overline{C} \rangle_F = \widehat{TC}(h(\overline{\pi}); C, \alpha, \beta) + \rho(n-m)\alpha_*$.*

*Proof.* We first recall the definition of $\overline{C}$ given component-wise in Eq. (8) by

$$\overline{C}_{ij} = \begin{cases} C_{ij}, & \text{if } i \leq m \text{ and } C_{ij} \leq \rho(\alpha_i + \beta_j) \\ \rho(\alpha_i + \beta_j), & \text{if } i \leq m \text{ and } C_{ij} > \rho(\alpha_i + \beta_j) \\ \rho(\alpha_* + \beta_j), & \text{if } i > m \end{cases}$$

We use case-by-case definition to decompose the Frobenius inner product. We now define index sets of rows and columns which indicate where $\overline{\pi}$ takes a value of 1 that violates the conditions for setting $\overline{C}_{ij}$, that is

$$\mathcal{I} = \{(i,j) : 1 \leq i \leq m, 1 \leq j \leq n, \text{ such that } \overline{\pi}_{ij} = 1 \text{ and } C_{ij} \leq \rho(\alpha_i + \beta_j)\}$$
$$\mathcal{J} = \{(i,j) : 1 \leq i \leq m, 1 \leq j \leq n, \text{ such that } \overline{\pi}_{ij} = 1 \text{ and } C_{ij} > \rho(\alpha_i + \beta_j)\}$$
$$\mathcal{K} = \{(i,j) : m < i \leq n, 1 \leq j \leq n, \text{ such that } \overline{\pi}_{ij} = 1\}$$

and we note that the sets are disjoint within $\{1, \ldots, n\} \times \{1, \ldots, n\}$. We now decompose the inner product as follows

$$\langle \overline{\pi}, \overline{C} \rangle = \sum_{(i,j) \in \mathcal{I}} C_{ij} h(\overline{\pi})_{ij} + \rho \left( \sum_{(i,j) \in \mathcal{J}} (\alpha_i + \beta_j)\overline{\pi}_{ij} + \sum_{(i,j) \in \mathcal{K}} (\alpha_\star + \beta_j)\overline{\pi}_{ij} \right)$$

where we have immediately used the fact that $\overline{\pi}_{ij} = h(\overline{\pi})_{ij}$ for all $(i,j) \in \mathcal{I}$. Note furthermore that $\mathcal{I}$ is exactly the index set where $h(\overline{\pi})$ is non-zero, hence that

$$\sum_{(i,j) \in \mathcal{I}} C_{ij} h(\overline{\pi})_{ij} = \langle C, h(\overline{\pi}) \rangle_F.$$

We recall the notation $h(\overline{\pi})_1$ that denotes the first marginal given by $(h(\overline{\pi})_1)_i = \sum_{j=1}^n h(\overline{\pi})_{ij}$. For any fixed row index $i$ where for some $j$ the pair $(i,j) \in \mathcal{J}$, then the the marginal vectors have values

$(\overline{\pi}_1)_i = 1$ whereas $(h(\overline{\pi})_1)_i = 0$. On the other hand, if for fixed $i$ if there is no pair $(i, j) \in \mathcal{J}$, then the marginal is given by $(h(\overline{\pi})_1)_i = 1$ but it does not contribute to the sum, and we see that we have

$$\sum_{(i,j)\in\mathcal{J}} \alpha_i \overline{\pi}_{ij} = \sum_{i=1}^{m} (1 - (h(\overline{\pi})_1)_i)\alpha_i.$$

We have a similar logic for the column sums, with the exception that some of the marginal mass is accounted for in the index set $\mathcal{K}$, so similar to above we have that

$$\sum_{(i,j)\in\mathcal{J}} \beta_j \overline{\pi}_{ij} + \sum_{(i,j)\in\mathcal{K}} \beta_j \overline{\pi}_{ij} = \sum_{j=1}^{n} (1 - (h(\overline{\pi})_2)_j)\beta_j.$$

Finally, as $\overline{\pi}$ is a complete permutation, we know that each row $m < i \le n$ has some index $j$ for which $\overline{\pi}_{ij} = 1$ and hence

$$\sum_{(i,j)\in\mathcal{K}} \alpha_\star \overline{\pi}_{ij} = (n - m)\alpha_\star.$$

All put together this yields

$$\langle \overline{\pi}, \overline{C} \rangle = \langle C, h(\overline{\pi}) \rangle_F + \rho \left( \sum_{i=1}^{m} \alpha_i (1 - (h(\overline{\pi})_1)_i) + \sum_{j=1}^{n} \beta_j (1 - (h(\overline{\pi})_2)_j) + (n - m)\alpha_\star \right)$$

$$= \langle C, h(\overline{\pi}) \rangle_F + \rho \left( \langle \alpha, \mathbf{1}_m - h(\overline{\pi})_1 \rangle + \langle \beta, \mathbf{1}_n - h(\overline{\pi})_2 \rangle \right) + \rho(n - m)\alpha_\star$$

$$= \widehat{TC}(h(\overline{\pi}); C, \alpha, \beta) + \rho(n - m)\alpha_\star.$$

$\square$

Before we prove Theorem 6.2, we prove the following two technical lemmas which will be used to prove Theorem 6.2.

**Lemma A.3.** *Let $\pi^* \in \mathcal{M}$ be a solution for the partial graph matching problem, that is $\pi^\star \in \arg\min_{\pi \in \mathcal{M}} \widehat{TC}(\pi; C, \alpha, \beta)$. For any $1 \le p \le m$ and any $1 \le q \le n$,*

$$((\pi_1^*)_p = 0) \text{ and } ((\pi_2^*)_q = 0) \implies C_{pq} \ge \rho(\alpha_p + \beta_q).$$

*Proof.* We prove this by contradiction and assume there exist $p, q$ such that $(\pi_1^*)_p = 0$, $(\pi_2^*)_q = 0$, and $C_{pq} < \rho(\alpha_p + \beta_q)$. As $(\pi_1^*)_p = 0$ and $(\pi_2^*)_q = 0$, $\widehat{TC}(\pi^*; C, \alpha, \beta)$ can be expanded as

$$\widehat{TC}(\pi^*; C, \alpha, \beta) = \sum_{\substack{1\le i\le m \\ i\neq p}} \sum_{\substack{1\le j\le n \\ j\neq q}} \pi_{ij}^* C_{ij} + \rho \left( \sum_{\substack{1\le i\le m \\ i\neq p}} \alpha_i (1 - (\pi_1^*)_i) + \alpha_p \right)$$

$$+ \rho \left( \sum_{\substack{1\le j\le n \\ j\neq q}} \beta_j (1 - (\pi_2^*)_j) + \beta_q \right)$$

Now we consider a plan $\pi' \in \mathcal{M}$ such that $\pi'_{pq} = 1$, but otherwise $\pi'_{ij} = \pi_{ij}^*$ for all other $1 \le i \le m$ and $1 \le j \le n$. Note that this is possible as mass of $p$ and $q$ is not transported in $\pi^*$. Based on marginal constraints, we know that $\pi'_{pq} = 1$ implies that $\pi'_{pj} = 0$ for all $j \neq q$ and $\pi'_{iq} = 0$ for all $i \neq p$. Therefore, $\widehat{TC}(\pi'; C, \alpha, \beta)$ can be rewritten as,

$$\widehat{TC}(\pi'; C, \alpha, \beta) = \sum_{\substack{1\le i\le m \\ i\neq p}} \sum_{\substack{1\le j\le n \\ j\neq q}} \pi'_{ij} C_{ij} + C_{pq} + \rho \left( \sum_{\substack{1\le i\le m \\ i\neq p}} \alpha_i (1 - (\pi'_1)_i) \right)$$

$$+ \rho \left( \sum_{\substack{1\le j\le n \\ j\neq q}} \beta_j (1 - (\pi'_2)_j) \right)$$

Thus we see that the difference in the objective between these two plans is $\widehat{TC}(\pi';C,\alpha,\beta) - \widehat{TC}(\pi^*;C,\alpha,\beta) = C_{pq} - \rho(\alpha_p + \beta_q)$, and finally as $C_{pq} < \rho(\alpha_p + \beta_q)$ we have

$$\widehat{TC}(\pi';C,\alpha,\beta) < \widehat{TC}(\pi^*;C,\alpha,\beta).$$

However, this is a contradiction as $\pi^\star \in \arg\min_{\pi\in\mathcal{M}} \widehat{TC}(\pi;C,\alpha,\beta)$. Therefore, the assumption is wrong. Thus, for any $\pi^\star \in \arg\min_{\pi\in\mathcal{M}} \widehat{TC}(\pi;C,\alpha,\beta)$, $(\pi_1^*)_p = 0$ and $(\pi_2^*)_q = 0$ implies that $C_{pq} \geq \rho(\alpha_p + \beta_q)$. $\qquad\square$

**Lemma A.4.** *Let $\pi^* \in \mathcal{M}$ be a solution for the partial graph matching problem, that is $\pi^\star \in \arg\min_{\pi\in\mathcal{M}} \widehat{TC}(\pi;C,\alpha,\beta)$. Then, there exists $\overline{\pi} \in \mathcal{P}_n$ such that $\langle \overline{\pi}, \overline{C} \rangle_F = \widehat{TC}(\pi^\star;C,\alpha,\beta) + \rho(n-m)\alpha_*$.*

*Proof.* Let us consider some $\pi \in \arg\min_{\pi\in\mathcal{M}} \widehat{TC}(\pi;C,\alpha,\beta)$. Our objective is to show that we can find $\pi' \in \mathcal{P}_n$ such that $\widehat{TC}(\alpha,\beta,\pi,C) + (n-m)\alpha^* = \langle \pi', \overline{C} \rangle$ where $\overline{C} \in \mathbb{R}^{n\times n}$ is defined as Eq. (8)

First, we extend $\pi$ to $\pi' \in \{0,1\}^{n\times n}$ by padding with zeros, defining it as

$$\pi'_{ij} = \begin{cases} \pi_{ij}, & \text{if } i \leq m \\ 0 & \text{if } i > m. \end{cases}$$

The idea of the proof that follows is that we define a permutation matrix $\overline{\pi} \in \mathcal{P}_n$ that is equal to $\pi$ where it is 1, but fills in the unmatched nodes with some arbitrary permutation. That is, we assume a decomposition s.t. for all $1 \leq i, j \leq n$

$$\overline{\pi}_{ij} = \pi'_{ij} + \hat{\pi}_{ij}, \tag{15}$$

with $\pi'_{ij} = 1$ if $\pi_{ij} = 1$, and we note that the marginals satisfy $\overline{\pi}_1 = \pi'_1 + \hat{\pi}_1 = \mathbf{1}_n$ and $\overline{\pi}_2 = \pi'_2 + \hat{\pi}_2 = \mathbf{1}_n$. Now the total cost function Eq. (7) can be rewritten as

$$\widehat{TC}(\pi;C,\alpha,\beta) = \langle \pi, C \rangle_F + \rho\left( \sum_{i=1}^{m} \alpha_i(\hat{\pi}_1)_i + \sum_{j=1}^{n} \beta_j(\hat{\pi}_2)_j \right)$$

By the definition of $\pi'_{ij}$, we know that for all $i > m$, $\pi'_{ij} = 0$. Therefore, for all $i > m$, $\hat{\pi}_1 = 1$. Thus, we have,

$$\sum_{i=m+1}^{n} (\hat{\pi}_1)_i = (n-m)$$

Therefore, we can derive the following,

$$\widehat{TC}(\pi;C,\alpha,\beta) + \rho(n-m)\alpha_* = \langle \pi, C \rangle_F + \rho\left( \sum_{i=1}^{m} \alpha_i(\hat{\pi}_1)_i + \sum_{j=1}^{n} \beta_j(\hat{\pi}_2)_j + \sum_{i=m+1}^{n} (\hat{\pi}_1)_i\alpha_* \right)$$

$$= \sum_{\substack{1\leq i\leq m \\ 1\leq j\leq n}} \pi'_{ij}C_{ij} + \sum_{\substack{1\leq i\leq m \\ 1\leq j\leq n}} \rho(\alpha_i + \beta_j)\hat{\pi}_{ij} + \sum_{\substack{m< i\leq n \\ 1\leq j\leq n}} \rho(\beta_j + \alpha_*)\hat{\pi}_{ij}$$

$$\tag{16}$$

The contrapositive of Theorem 5.1 tells us that as $\pi$ is optimal, $\pi_{ij} = 1$ implies that $C_{ij} \leq \rho(\alpha_i + \beta_j)$. By definition of $\pi'$, we know that, $\pi'_{ij} = 1$ if and only $\pi_{ij} = 1$, so hence $\pi'_{ij} = 1$ also implies that $C_{ij} \leq \rho(\alpha_i + \beta_j)$. Thus, based on the definition of $\overline{C}$ in Eq. (8), we can deduce that

$$\sum_{\substack{1\leq i\leq m \\ 1\leq j\leq n}} \pi_{ij}C_{ij} = \sum_{\substack{1\leq i\leq m \\ 1\leq j\leq n}} \pi'_{ij}C_{ij} = \sum_{\substack{1\leq i\leq m \\ 1\leq j\leq n}} \pi'_{ij}\overline{C}_{ij} \tag{17}$$

From Eq. (15), we know that, for any $1 \leq i, j \leq n$, $\hat{\pi}_{ij} = 1$ if and only if $(\pi'_1)_i = 0$ and $(\pi'_2)_j = 0$. Moreover, from the definition of $\pi'$, for any $1 \leq i \leq m$ and $1 \leq j \leq n$, $(\pi'_1)_i = 0$ and $(\pi'_2)_j = 0$ if and only if $(\pi_1)_i = 0$ and $(\pi_2)_j = 0$. And, from Lemma A.3, we know that $(\pi_1)_i = 0$ and $(\pi_2)_j = 0$ implies that $C_{ij} \geq \rho(\alpha_i + \beta_j)$. Therefore, for any $1 \leq i \leq m$ and $1 \leq j \leq n$, $\hat{\pi}_{ij} = 1$ implies that $C_{ij} \geq \rho(\alpha_i + \beta_j)$ hence that $\overline{C}_{ij} = \rho(\alpha_i + \beta_j)$. Thus, we can deduce,

$$\sum_{\substack{1 \leq i \leq m \\ 1 \leq j \leq n}} \rho(\alpha_i + \beta_j)\hat{\pi}_{ij} = \sum_{\substack{1 \leq i \leq m \\ 1 \leq j \leq n}} \hat{\pi}_{ij}\overline{C}_{ij} \tag{18}$$

From definition of $\overline{C}$, we know that for all $i > m, \overline{C}_{ij} = \rho(\alpha_* + \beta_j)$. Therefore, we have,

$$\sum_{\substack{m < i \leq n \\ 1 \leq j \leq n}} \rho(\beta_j + \alpha_*)\hat{\pi}_{ij} = \sum_{\substack{m < i \leq n \\ 1 \leq j \leq n}} \hat{\pi}_{ij}\overline{C} \tag{19}$$

From Eqs. (17) to (19), and using the fact that $\pi'_{ij} = 0$ for $m < i \leq n$, we can rewrite Eq. (16) as

$$\begin{aligned}
\widehat{TC}(\pi; C, \alpha, \beta) + \rho(n - m)\alpha_* &= \sum_{\substack{1 \leq i \leq m \\ 1 \leq j \leq n}} \pi'_{ij}\overline{C}_{ij} + \sum_{\substack{1 \leq i \leq m \\ 1 \leq j \leq n}} \hat{\pi}_{ij}\overline{C}_{ij} + \sum_{\substack{m < i \leq n \\ 1 \leq j \leq n}} \hat{\pi}_{ij}\overline{C}_{ij} \\
&= \sum_{\substack{1 \leq i \leq n \\ 1 \leq j \leq n}} \pi'_{ij}\overline{C}_{ij} + \sum_{\substack{1 \leq i \leq n \\ 1 \leq j \leq n}} \hat{\pi}_{ij}\overline{C}_{ij} \\
&= \langle \overline{\pi}, \overline{C} \rangle_F \tag{20}
\end{aligned}$$

□

**Theorem 6.2** (Optimal Solution). *If $\overline{\pi}^\star \in \arg\min_{\overline{\pi} \in \mathcal{P}_n} \langle \overline{\pi}, \overline{C} \rangle_F$, then $\pi^\star = h(\overline{\pi}^\star)$ is a solution of partial graph matching problem, that is $\pi^\star \in \arg\min_{\pi \in \mathcal{M}} \widehat{TC}(\pi; C, \alpha, \beta)$.*

*Proof.* Let $\overline{\pi}^\star \in \arg\min_{\overline{\pi} \in \mathcal{P}} \langle \overline{\pi}, \overline{C} \rangle_F$ be any optimal solution of Eq. (9) and $\pi^\star \in \arg\min_{\pi \in \mathcal{M}} \widehat{TC}(\pi; C, \alpha, \beta)$ be any optimal solution of Eq. (5)

From Lemma A.4, we know that, there exists a $\overline{\pi} \in \mathcal{P}_n$ s.t. $\widehat{TC}(\pi^\star; C, \alpha, \beta) + (n - m)\alpha_* = \langle \overline{\pi}, \overline{C} \rangle$. Therefore,

$$\langle \overline{\pi}^*, \overline{C} \rangle \leq \langle \overline{\pi}, \overline{C} \rangle = \widehat{TC}(\pi^\star; C, \alpha, \beta) + \rho(n - m)\alpha_* \tag{21}$$

From Lemma 6.1, we have $h(\overline{\pi}^*) \in \mathcal{M}$ as defined in Eq. (10) from $\overline{\pi}^*$ such that the following condition holds,

$$\widehat{TC}(h(\overline{\pi}^*); C, \alpha, \beta) + (n - m)\alpha_* = \langle \overline{\pi}^*, \overline{C} \rangle \tag{22}$$

Finally, from Eq. (21) and Eq. (22),

$$\widehat{TC}(h(\overline{\pi}^*); C, \alpha, \beta) + \rho(n - m)\alpha_* \leq \widehat{TC}(\pi^\star; C, \alpha, \beta) + \rho(n - m)\alpha_*$$

and as $\pi^\star$ is already a minimizer of $\widehat{TC}$, implies that $\widehat{TC}(h(\overline{\pi}^*); C, \alpha, \beta) = \widehat{TC}(\pi^\star; C, \alpha, \beta)$, which indicates that $h(\overline{\pi}^*) \in \arg\min_{\pi \in \mathcal{M}} \widehat{TC}(\pi; C, \alpha, \beta)$.

Thus, from any $\overline{\pi}^* \in \arg\min_{\overline{\pi}' \in \mathcal{P}_n} \langle \overline{\pi}, \overline{C} \rangle_F$, which is an optimal solution of Eq. (9), it is possible to derive $h(\overline{\pi}^*) \in \arg\min_{\pi \in \mathcal{M}} \widehat{TC}(\pi; C, \alpha, \beta)$, which solves Eq. (6)). Thus, the proof is complete.

□

## B ADDITIONAL DETAILS ON EXPERIMENTS

**Discussion on PascalVOC and PPI Network Matching** As shown in Table 3, OPGM and OPGM-rs achieve lower mean matching F1-scores compared to the AFAT-U (GCAN) and AFAT-I (GCAN) models. This is mainly due to the poor performance of OPGM-rs in the *table* and *sofa* classes, which are known to have ambiguous and poor annotations. Notably, these classes were excluded from the SPair-71K dataset for the same reason (Rolínek et al., 2020).

| GM Network | PMH | aero | bike | bird | boat | bottle | bus | car | cat | chair | cow | table | dog | horse | mbike | person | plant | sheep | sofa | train | tv | mean |
|---|---|---|---|---|---|---|---|---|---|---|---|---|---|---|---|---|---|---|---|---|---|---|
| NGM-v2 | dummy | 44.7 | 61.9 | 57.1 | 41.9 | 83.9 | 63.9 | 54.1 | 60.8 | 40.5 | 64.2 | 36.2 | 60.6 | 60.8 | 61.9 | 48.7 | 91.2 | 56.2 | 37.4 | 63.2 | 82.2 | 58.6±0.5 |
| NGM-v2 | AFAT-U | 45.7 | 67.7 | 57.3 | 44.9 | 90.1 | 65.5 | 49.9 | 59.3 | 44.0 | 62.0 | 54.9 | 58.4 | 58.6 | 63.8 | 45.9 | 94.8 | 50.9 | 37.3 | 74.2 | 82.8 | 60.2±0.4 |
| NGM-v2 | AFAT-I | 45.0 | 67.3 | 55.9 | 45.6 | 90.3 | 64.6 | 48.7 | 58.0 | 44.7 | 60.2 | 54.8 | 57.2 | 57.5 | 63.4 | 45.2 | 95.3 | 49.3 | 41.6 | 73.6 | 82.4 | 59.9±0.3 |
| GCAN | ILP | 46.3 | 67.7 | 57.4 | 45.0 | 87.1 | 64.8 | 57.5 | 61.2 | 40.8 | 61.6 | 37.3 | 59.9 | 59.2 | 64.6 | 49.7 | 95.1 | 54.5 | 28.5 | 77.9 | 83.1 | 59.7±0.3 |
| GCAN | AFAT-U | 47.1 | 70.8 | 58.1 | 45.8 | 90.8 | 66.5 | 49.6 | 58.8 | 50.6 | 64.6 | 47.2 | 60.5 | 62.3 | 65.7 | 46.3 | 95.4 | 52.7 | 47.4 | 74.2 | 83.8 | 62.0±0.2 |
| GCAN | AFAT-I | 46.1 | 69.9 | 56.1 | 46.6 | 90.7 | 66.1 | 48.1 | 57.9 | 49.9 | 63.9 | 50.4 | 59.0 | 61.6 | 65.0 | 44.7 | 95.5 | 50.9 | 49.2 | 74.0 | 83.8 | 61.6±0.3 |
| GCAN | OPGM | 46.5 | 70.1 | 55.8 | 47.1 | 89.5 | 62.4 | 46 | 60.8 | 48.9 | 63.4 | 46.2 | 58.9 | 60.2 | 67.7 | 47.7 | 94.9 | 51.5 | 42 | 73.1 | 82.9 | 60.8 ±0.4 |
| GCAN | OPGM-rs | 47.1 | 71.5 | 57.5 | 47.9 | 89.6 | 64.1 | 46.8 | 62.3 | 50.0 | 65.4 | 33.2 | 59.4 | 61.2 | 68.7 | 48.5 | 95.6 | 53.4 | 39.8 | 74.1 | 83.1 | 61.0 ±0.2 |

Table 3: Matching F1-score on Pascal VOC Keypoint. The best results are colored in **black** and the second best are in **blue**

.

| Method | Yeast 5% | Yeast 10% | Yeast 15% | Yeast 20% | Yeast 25% |
|---|---|---|---|---|---|
| SIGMA | 84.7±0.4 | 68.8 ±2.5 | 57.4±1.1 | 46.7 ±2.3 | 41.4 ±1.7 |
| StableGM | 86.1 ± 0.9 | 75.6 ±0.8 | 67.9 ± 1.1 | 63.2 ±0.9 | 57 ± 0.6 |
| OPGM (ours) | 87.8 ± 0.3 | 80 ±0.4 | 71.9 ±0.9 | 66.9 ±1.0 | 58.8 ±0.8 |
| OPGM-rs (ours) | 88.3 ±0.5 | 79.8 ±0.6 | 71.7 ±0.8 | 66.2 ±0.9 | 57.7 ±1.0 |

Table 4: Node correctness (%) results on the PPI dataset. The best results are colored in **black** and the second best are in **blue**

.

In PPI network matching, as shown in Table 4, OPGM-rs generally outperforms OPGM at lower noise levels, but OPGM demonstrates better performance as noise increases. This behavior can be attributed to the learnable cost matrix $C$ in OPGM-rs, which is utilized during training. When inputs are noisy, the information in $C$ may also become noisy, and simultaneously learning $\alpha$ and $\beta$ alongside $C$ can negatively impact the training process under high noise levels. These observations indicate that OPGM-rs is more sensitive to poor or ambiguous annotations and increased noise in the data, which adversely affects its performance.

**Discussion on efficiency analysis** As detailed in Section 8, all the models analyzed in Table 2(right) use the same neural architecture (Jiang et al., 2022b) to compute the cross-graph node-to-node affinity matrix and apply Sinkhorn normalization to obtain the doubly stochastic affinity matrix.

Our model, OPGM-rs, solves the partial graph matching problem as described in Section 6 by solving a linear sum assignment problem (LAP). We use SciPy's LAP solver (Virtanen et al., 2020), which is computationally efficient. In contrast, GCAN (Jiang et al., 2022b) formulates partial graph matching as an integer linear programming (ILP) problem, which is solved using Google's OR-tools (Perron & Furnon) in (Wang et al., 2023). Solving an ILP problem is generally more computationally expensive than solving a linear sum assignment problem using LAP solvers.

AFAT-I and AFAT-U models introduce additional complexities by employing separate neural modules to estimate the number of matchings between graphs. This process includes solving an entropy regularized optimal transport problem, followed by the GreedyTopK algorithm (Wang et al., 2023), which is based on the Hungarian algorithm, to compute the final matching. These extra steps increase the average inference time of AFAT-I and AFAT-U compared to OPGM-rs.

All experiments on efficiency analysis were conducted on a Linux server equipped with an Intel Xeon W-2175 2.50GHz processor (28 cores), an NVIDIA RTX A6000 GPU, and 512GB of main memory.

**Hyperparameters** For the image keypoint matching task, the hyperparameters of OPGM-rs are searched within the following ranges: $\rho \in \{0.1, 0.2, 0.3, 0.4, 0.5\}$, $\lambda \in \{0.01, 0.2, 0.3, 0.4, 0.5, 0.6, 0.8, 1.0\}$, learning rate $\in \{0.001, 0.002\}$, VGG16 backbone learning rate $\in \{0.0001\}$, batch size $\in \{4, 8\}$, and number of epochs $\in \{15, 20, 25\}$. To select $\rho$, we perform a grid search from 0.1 to 0.5 with a step size of 0.1. We use the Adam algorithm (Diederik, 2014) as our optimizer, and the initial learning rate decays by a factor of 0.5 after every two epochs.

For the PPI network matching task, the hyperparameters of OPGM-rs are searched within the following ranges: $\rho = 10^{11}$, $\lambda \in \{0.25, 0.5, 0.75, 1.0\}$, learning rate $\in \{0.0001, 0.0002\}$, and number of epochs $= 100$. The Adam algorithm (Diederik, 2014) is also used as the optimizer.

**Failure mode analysis** The feasibility of matching a pair of nodes $i$ and $j$ is influenced by their cost $C_{ij}$, the matching biases $\alpha_i$ and $\beta_j$, and the hyperparameter $\rho$. A specific failure mode we observed

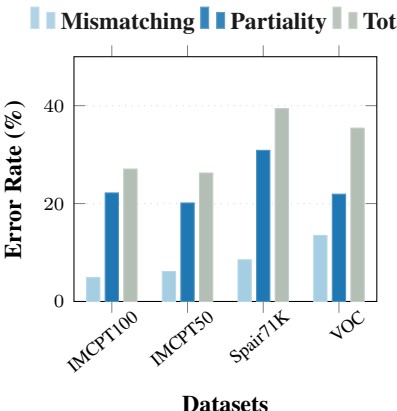

Figure 3: Failure mode analysis across four different datasets.

is when node pairs that should be matched according to the ground truth are deemed infeasible by the optimization objective in Eq. (9) (as described in Theorem 5.1). Consequently, these nodes remain unmatched in the final solution. This failure mode typically arises when the graph matching network $NN_\theta$ assigns an excessively low affinity (resulting in a high matching cost) to pairs of nodes that should otherwise be matched.

We quantify the error by analyzing:

- *Partiality error:* The percentage of true matches (as per the ground truth) that fail to meet the feasibility condition outlined in Theorem 5.1. This error reflects mismatches caused by incorrect partiality identification involving matching biases.

- *Mismatching error:* The percentage of true matches that are not identified as true matches even though they meet the feasibility condition.

Figure 3 demonstrates the partiality error, mismatching error and total error related to this failure mode across four different datasets. The total error is equal to the summation of partiality error and the mismatching error.

