# OpenReview forum: "Learning Partial Graph Matching via Optimal Partial Transport"
_ICLR.cc/2025/Conference — ICLR 2025 Poster_

### Official Review · Reviewer_KhhU · 2024-10-17

**Soundness:** 2
**Presentation:** 3
**Contribution:** 1
**Rating:** 6
**Confidence:** 4

**Summary:**

The paper formulated the partial graph matching problem as an optimal partial transport problem, enabling partial assignments (not have to be a bijective mapping) between a source graph and the target graph.The linear solver Hungarian algorithm is adopted to the LAP objective.

**Strengths:**

1. Formulating partial graph matching problem into an optimal transport problem is natural.
2. The objective is a linear assignment problem (LAP) and easy to be optimized.
3.  Both qualitative and quantitative results on several commonly used datasets valid the effectiveness of the proposed formulation.

**Weaknesses:**

1. The main contribution of the paper is to formulate the partial graph matching problem into an optimal partial transport problem, which is not new [1,2] and is a common relaxation of the original quadratic objective of graph matching involving both vertices and edges constraints (into a linear assignment problem).
2. Line 065 is not 100% accurate since there are some efficient algorithms (e.g. [3] is quite intuitive and easy to implement as the k-LAP is transformed into a standard LAP that can be efficiently solved by the Hungarian algorithm) targeting towards k-assignment problem. As such, manually filtering out infeasible assignments with a hard threshold in Theorem 5.1 is not necessary.
3. Some important references are missing [4,5].
4. The objective of Eq(9) is almost equivalent to the first order term of Koopmans-Beckmann’s QAP with partial matching constraints in Eq(10), and the network design is the same as GCAN, together with the common linear solver Hungarian algorithm making the contributions both network design and optimization insight a bit marginal. In fact, one promising and more insightful way for the paper to stand out among others is to explicit formulate graph edges into optimal transport objective or constraints rather only implicitly involved into feature aggregation by GCN.

To conclude, the paper proposed an optimal transport objective with partial matching constraints for partial graph matching problem. However, I haven't seen any special design for the partial graph matching nor optimal transport.



[1] Pan et al. Subgraph Matching via Partial Optimal Transport. arxiv, 2024
[2] Xu et al. Gromov-Wasserstein Learning for Graph Matching and Node Embedding. ICML, 2019.
[3] Volgenant et al. Solving the k-cardinality assignment problem by transformation. European Journal of Operational Research, 2004.
[4] Fey et al. Deep Graph Matching Consensus. ICLR, 2020.
[5] Gao et al. Deep graph matching under quadratic constraint. CVPR, 2021.

**Questions:**

Theorem 5.1 about filtering infeasible assignments with a hard threshold seems to be not necessary even without a k-LAP algorithm. Since the proposed method will output an optimal partial matching in anyway, why it needs to filter out some infeasible assignments at the very beginning?

---

> ### Author Response · Authors · 2024-11-25
> **Responses for Reviewer KhhU's Comments**
>
> **W1:** *The main contribution of the paper is to formulate the partial graph matching problem into an optimal partial transport problem, which is not new [1,2] and is a common relaxation of the original quadratic objective of graph matching involving both vertices and edges constraints (into a linear assignment problem).*
>
>
> **Response:** We thank the reviewer for giving the opportunity to clarify the novelty of our work.
> While there are connections to [1] and [2], we believe our methodology is fundamentally different in critical ways.
>
> In [1], optimal partial transport is applied to subgraph matching with predefined mass, which works because the number of nodes in the query graph is known. However, in our partial graph matching setting, the number of matched nodes is unknown as both graphs may have unmatched nodes. Formulations like [1] that rely on predefined mass constraints are therefore not applicable. Unlike [1], our method dynamically determines the optimal mass during optimization, making it suitable for partial graph matching.
>
> In [2], the authors address graphs with different degrees by normalizing weights, casting the problem into a quadratic Gromov-Wasserstein (GW) framework that incorporates node embeddings and edge weights. However, the GW problem is non-convex, computationally expensive, and prone to local minima. Moreover, the resulting transport matrices are not guaranteed to be assignment matrices, as a binary (but not bijective) matrix is derived, allowing multiple source nodes to match the same target node.
>
> In contrast, our approach does offer these missing guarantees.
>
> - Our formulation guarantees a partial assignment adhering to the matching constraints of partial graph matching and that is optimal with respect to our objective. (In [2], the final assignment is not optimal w.r.t. the proposed objective given in [2], and both [1,2] consider different matching constraints than a partial assignment.)
> - Our optimization problem is linear and can be guaranteed to be exactly solved to find an optimal partial assignment within a cubic worst-case complexity. Both [1] and [2] consider quadratic objectives, and quadratic objectives have higher computational complexity than linear objectives.
> - The subset of matching nodes within the source and target graphs is determined directly within the optimization. (In [1], the number of nodes to be matched from the smaller graph is known, and in [2], it is assumed each node in the smaller graph will be matched for some node in the other graph.)
>
>  We have these guarantees as we embed the partial optimal transport problem in a balanced transport problem given in Eq. (9), and utilizing the results of Bai and Caffarelli \& McCann to show existence of solutions and equivalence.
> Furthermore we can use a linear objective as node embeddings are capable of capturing edge information, which we consider to be a reasonable tradeoff against including edge weights in a final quadratic Gromov-Wasserstein objective, especially given the computational cost of GW.
>
> [1] Pan et al. Subgraph Matching via Partial Optimal Transport. arxiv, 2024.
>
> [2] Xu et al. Gromov-Wasserstein Learning for Graph Matching and Node Embedding. ICML, 2019.
>
> Note : Please note that owing to the limitations on characters, we have provided our responses to the remaining questions of the reviewer as replies to this comment.

---

> ### Author Response · Authors · 2024-11-25
> **Responses for Reviewer KhhU's Comments**
>
> **W2:** *Line 065 is not 100\% accurate since there are some efficient algorithms (e.g. [3] is quite intuitive and easy to implement as the k-LAP is transformed into a standard LAP that can be efficiently solved by the Hungarian algorithm) targeting towards k-assignment problem. As such, manually filtering out infeasible assignments with a hard threshold in Theorem 5.1 is not necessary.*
>
> **Response:** We would like to clarify a misunderstanding: our method *does not* manually filter out infeasible assignments using a hard threshold as described in Theorem 5.1. The feasibility threshold in Theorem 5.1 is derived from the optimization objective and is an inherent property of the model, not a manually imposed constraint.
>
>
> We agree with the reviewer that k-LAP can be transformed into a standard LAP and solved using the Hungarian algorithm. However, in partial graph matching, the number of nodes \(k\) to be matched is unknown. Solving partial graph matching as a $k$-assignment problem first requires estimating \(k\), which is non-trivial. For instance, [6] employs a separate module to estimate \(k\) by solving an entropy-regularized optimal transport problem, followed by solving a $k$-assignment problem using the GreedyTopK algorithm. This two-step approach increases computational complexity and risks error propagation from incorrect \(k\)-estimates.
>
> In contrast, our method determines the number of nodes to be matched *directly within* the optimal transport optimization, eliminating the need for separate estimation. This makes our approach both more efficient and robust compared to methods that treat partial graph matching as a $k$-assignment problem (see experiments in Table 2 for details).
>
> **W3:** *Some important references are missing [4,5].*
>
> **Response:** We thank the reviewer for pointing out some references related to deep graph matching. We have added these references into the revised manuscript.
>
> **W4:** *The objective of Eq(9) is almost equivalent to the first order term of Koopmans-Beckmann’s QAP with partial matching constraints in Eq(10), and the network design is the same as GCAN, together with the common linear solver Hungarian algorithm making the contributions both network design and optimization insight a bit marginal. In fact, one promising and more insightful way for the paper to stand out among others is to explicit formulate graph edges into optimal transport objective or constraints rather only implicitly involved into feature aggregation by GCN.*
>
>
>
> **Response:**
>  Eq.(9) is a linear objective equivalent to the linear term in Koopmans-Beckmann’s formulation. However, this equivalence can also be drawn for many linear problems, such as balanced Optimal Transport itself. What sets our method apart is its ability to reduce the difficult problem of determining *partial* assignments into the relatively simpler problem of linear full-assignment. This is the crux of our approach, and stands in contrast with other methods where the partial matching problem is either solved with a quadratic objective or a linear objective with a partial term that must be optimised concurrently with the assignment, thereby increasing complexity.
>
> The ability to exactly solve the partial graph matching problem given in Eq. (6) within a cubic worst case time complexity utilizing the Hungarian algorithm is only possible because of the theoretical connection between Eq.(6) and Eq.(9) that is shown in this study (Lemma 6.1, Theorem 6.2), which we consider a significant result.
>
> **Q1:** *Theorem 5.1 about filtering infeasible assignments with a hard threshold seems to be not necessary even without a k-LAP algorithm. Since the proposed method will output an optimal partial matching in anyway, why it needs to filter out some infeasible assignments at the very beginning?*
>
> **Response:** This is a good observation. Indeed, the feasibility threshold in Theorem 5.1 is not a manually imposed hard threshold. Rather, it is an inherent property of the optimization objective itself. The model inherently rule out infeasible assignments during the optimization process, without requiring any explicit or manual filtering.
>
> We recognize that the term "feasibility threshold" might have led to confusion. To address this, we have revised the terminology in the manuscript to clarify its role and avoid misunderstandings. Thank you for bringing this to our attention.
>
> [3] Volgenant et al. Solving the k-cardinality assignment problem by transformation. European Journal of Operational Research, 2004.
>
> [4] Fey et al. Deep Graph Matching Consensus. ICLR, 2020.
>
> [5] Gao et al. Deep graph matching under quadratic constraint. CVPR, 2021.
>
> [6]  Wang et al. Deep learning of partial graph matching via differentiable top-k, CVPR 2023.

---

> > ### Comment · Reviewer_KhhU · 2024-11-26
> >
> > I appreciate the authors' detailed response.
> > Thanks for the clarification, I have a follow up question regarding to " the number of nodes (k) to be matched is unknown". Since the optimization is formulated as a LAP as described in Eq(9) and is solved using the Hungarian algorithm, the number of nodes to be matched appears to be fixed to satisfy the definition of a permutation matrix, which is at least an injection from source nodes to target nodes. So why "the number of nodes (k) to be matched is unknown"?
> >
> > Best,
> > reviewer KhhU

---

> ### Author Response · Authors · 2024-11-27
>
> We thank the reviewer for providing the opportunity to further clarify our method. Yes, we agree that solving the linear assignment problem in Eq. (9) will return a permutation matrix which is effectively a full matching. However, it should be noted that an optimal solution for Eq. (9) itself is not an optimal solution for Eq. (6). Therefore, we would like to clarify that we do not consider a solution for Eq. (9) as a solution for the partial graph matching problem given in Eq. (6).
>
> In this work, we have shown that there is a closed-form mapping from an optimal solution of Eq. (9) to an optimal solution of Eq. (6), as discussed in Theorem 6.2 and Eq. (10). When this closed-form mapping is applied to a solution of Eq. (9), an optimal solution for Eq. (6) will be obtained (Theorem 6.2).
>
> In total, this means we use a solution of Eq. (9) to derive an optimal solution for Eq. (6). The objective defined in Eq. (6) is designed to dynamically balance between nodes to be matched versus unmatched, considering the cost of matching two nodes versus the cost of leaving those two nodes unmatched. When the closed-form mapping is applied, some of the matchings in the solution obtained by solving Eq. (9) will be discarded such that the final solution will be optimal with respect to Eq. (6).
> Thus, the final number of nodes $k$ to be matched is determined by the optimality with respect to the objective defined in Eq. (6).
>
> The point of using the linear assignment optimization in Eq. (9) to derive a solution for Eq. (6) is that it is easier to solve Eq. (9) than solving Eq. (6) directly.

---

> > ### Comment · Reviewer_KhhU · 2024-11-27
> >
> > I would like to thank again for authors' response. Sorry for my misunderstanding, now I see.
> > Please correct me if I was wrong, it appears that the closed-form mapping from Eq(8) to Eq(10) always results in a solution $h$ with at least $m$ matching, i.e. an injective mapping from source to target. Does the proposed method allow the partial graph matching, where only a subset of source points is matched to a subset of target points?
> >
> > Best, reviewer KhhU

---

> ### Author Response · Authors · 2024-11-28
>
> We thank the reviewer for providing us an opportunity to further clarify on the closed-form mapping discussed in our work. The proposed method allows partial graph matching, where only a subset of source points is matched to a subset of target points. Please find our detailed explanation below,
>
> First, we would like to clarify that when obtaining the final solution, we apply the closed-form mapping $h : \mathbb{R}^{n \times n} \to \mathbb{R}^{m \times n}$ (described in Eq.(10)) to a solution of Eq.(9), which is a $n \times n$ permutation matrix and obtain a $m \times n$ matrix (which we prove is a solution to Eq. (6) in Theorem 6.2).
>
> Let $\overline{\pi}^\star$ be any solution for Eq. (9). The closed-form mapping $h$ (which is a mapping from a $n \times n$ matrix to a $m \times n$ matrix), when applies to $\overline{\pi}^\star$, first effectively discards the last $(n-m)$ rows of $\overline{\pi}^\star$, that is related with $n-m$ dummy elements (See paragraph 2 of section 6 for further details). After discarding the last $(n-m)$ rows, the resulting matrix $\overline{\pi}^\star$[1:m][1:n] has exactly $m$ number of matches (as $\overline{\pi}^\star$ is a permutation matrix). However, it should be noted that the closed-form mapping $h$ (eq. (10)) further discards matches in $\overline{\pi}^\star$[1:m][1:n] that does not satisfy the condition $C_{ij} \leq \rho(\alpha_i+\beta_j)$. Therefore, after the overall process, the total number of matches in the resulting matrix  $h(\overline{\pi}^\star)$  will be *less than or equal* to $m$.
>
> It is worth noting that as shown in Eq. (10), the feasibility condition is checked based on the original cost matrix $C$ (not $\overline{C}$ discussed in Eq. (8)), and $C$ can have values that does not satisfy the condition $C_{ij} \leq \rho(\alpha_i+\beta_j)$. Thus, the final number of matchings that will prevail after applying closed-form mapping $h$ to a solution in Eq.(9) can be any integer value from 0 to m (including 0 and m).
>
> We believe our answer clarifies the reviewer's question and we are happy to provide further explanations if needed.

---

> > ### Comment · Reviewer_KhhU · 2024-11-28
> >
> > I would like to thank the authors for the detailed clarifications. I would like to raise my score to 6.
> >
> > Best, reviewer KhhU

---

> > > ### Author Response · Authors · 2024-11-28
> > >
> > > We sincerely appreciate the reviewer's detailed review of our manuscript and we thank the reviewer for raising the score.

---

### Official Review · Reviewer_aXCT · 2024-10-28

**Soundness:** 1
**Presentation:** 2
**Contribution:** 2
**Rating:** 5
**Confidence:** 4

**Summary:**

This paper introduces OPGM, which addresses the partial graph matching problem. It begins by establishing theoretical connections between optimal partial transport and the partial matching problem. The paper then presents a method that leverages this formulation to learn matching bias and a cost affinity matrix, improving the outcomes of partial graph matching. Experimental results on various datasets demonstrate that this method is comparable or better than existing approaches.

**Strengths:**

The motivation behind the study is clear, and the results presented looks convincing.

**Weaknesses:**

1. The connection between optimal transport and the graph matching problem is already well-established, making it seem redundant to re-establish this connection in the partial setting.

2. The use of the term “optimal” in the title may be misleading. A significant challenge in partial matching is setting thresholds ($\rho$, $\alpha$, $\beta$ in this paper) to eliminate redundant points. The paper assumes these thresholds are predefined, which undermines the rigor of claiming their results as “optimal”.

3. PascalVOC is still the most important benchmark for the graph matching community, including its result only in the appendix does not seem appropriate, especially when this method does not do better on this benchmark.

4. The paper omits two critical references [1,2], both of which have achieved better results on PascalVOC.

[1] Nurlanov et al., "Universe Points Representation Learning for Partial Multi-Graph Matching"

[2] Jiang et al., "Learning Universe Model for Partial Matching Networks over Multiple Graphs"

**Questions:**

1. How effective is the applied bias term in identifying non-matching points, and what are the false positive and negative rates? It would be important to quantify the effectiveness of the proposed scheme.

2. What would the performance of other methods look like if they simply applied a thresholding scheme on their cost matrix similar to this study? An ablation study would be appreciated.

3. How is the parameter $\rho$ chosen? Is it chosen by some heuristics? Fig. 2 (middle) suggests that $\rho$ values supporting better performance are quite narrow (0.35-0.4), and these values vary across different tasks. Determining this parameter is one of the most challenging aspects of partial matching, so it is essential to understand how to set $\rho$ without extensive trial and error.

---

> ### Author Response · Authors · 2024-11-25
> **Responses for Reviewer aXCT's Comments**
>
> **W1:** *The connection between optimal transport and the graph matching problem is already well-established, making it seem redundant to re-establish this connection in the partial setting.*
>
> **Response:** We thank the reviewer for giving the opportunity to clarify the novelty of our work.
>
> We agree with the reviewer that the connection between optimal transport and graph matching is already established [1]. However, such approaches are not directly applicable to partial graph matching, mainly due to the fact that partial graph matching is a more generalized version of the graph matching problem. While graph matching assumes each node in the smaller graph is matched with some node in the other graph, partial graph matching allows nodes from both graphs to remain unmatched, thus making graph matching a specific case of partial graph matching. We have detailed the limitations of using existing optimal transport-based graph matching techniques for partial graph matching below.
>
> It should be noted that solving graph matching as an optimal transport problem does not inherently enforce one-to-one matching constraints [1] (i.e., in graph matching, each node in one graph cannot be mapped to more than one node in the other graph. However, in a transportation plan, a mass corresponding to a node can be transported to several nodes in the other graph ). One approach to overcome this challenge is to consider graph matching as an optimal transport problem between two discrete distributions with uniform mass and equal cardinality (dummy elements will need to be added in case the cardinality of the two node sets is different), and then use the Hungarian algorithm to find the optimal plan. However, this approach would still map all nodes in the smaller graph to the other graph, making it suitable for graph matching, but not for partial graph matching. This limitation of existing optimal transport-based graph matching techniques is mainly due to the conservation of mass constraint in optimal transport formulations, which requires all mass from one distribution to be transported to the other distribution.
>
> However, in optimal partial transport, the mass conservation constraint is relaxed, and the optimal partial mass to be transported is decided within the optimization. This optimization objective shares a similarity to the partial graph matching problem, in the sense that partial graph matching seeks an optimal mapping where not all nodes are matched, while optimal partial transport identifies a plan where not all mass is transported. However, a key difference exists: partial graph matching requires the mapping between two sets to be an injective partial function (i.e., a partial assignment), whereas optimal partial transport does not (a transportation plan obtained by solving an optimal partial transport problem does not inherently adhere to the one-to-one matching constraints of the partial graph matching problem). In this work, we discuss how to bridge this key difference and provide an optimization objective to produce an optimal partial matching solution adhering to the one-to-one matching constraints of the partial graph matching problem. Our proposed mechanism is significantly more efficient compared to existing partial graph matching techniques (experimental details are available in Table 2 (right)).
>
> [1] H. Xu, D. Luo, H. Zha, and L. C. Duke, “Gromov-wasserstein learning for graph matching and node embedding,” in International conference on machine learning.PMLR, 2019, pp. 6932–6941.
>
> Note : Please note that owing to the limitations on characters, we have provided our responses to the remaining questions of the reviewer as replies to this comment.

---

> ### Author Response · Authors · 2024-11-25
> **Responses for Reviewer aXCT's Comments**
>
> **W2:** *The use of the term “optimal” in the title may be misleading. A significant challenge in partial matching is setting thresholds ($\rho,\alpha,\beta$ in this paper) to eliminate redundant points. The paper assumes these thresholds are predefined, which undermines the rigor of claiming their results as “optimal”.*
>
> **Response:** While $\rho$ is indeed a hyperparameter representing the unbalancedness in our optimization objective, the values for $\alpha, \beta$ are in fact learned in our method, not manually setting as a threshold to eliminate redundant points.
>
> The term "optimal" that was initially in the title referred to the mathematically optimal solution to a well-defined optimization problem, rather than suggesting globally optimal solution for all possible partial matching scenarios. However, we agree with the reviewer and revised the title of the manuscript.
>
> We would also like to clarify the concept of the feasibility threshold discussed in Theorem 5.1. The feasibility threshold is *a property of the optimization objective* defined in Eq. (6) and *does not necessarily result in the elimination of a node.* If the cost between two nodes is higher than their feasibility threshold, they are ruled out of matching with each other, but it is still possible for each of those two nodes to be matched with other nodes in the other graph. Finally, if the cost between two nodes are less than the feasibility threshold, it does not guarantee that the two nodes will be matched in the optimal solution of Eq. (6). We have revised the term "feasibility threshold" to "feasibility condition" in the revised manuscript.
>
> **W3:** *PascalVOC is still the most important benchmark for the graph matching community, including its result only in the appendix does not seem appropriate, especially when this method does not do better on this benchmark.*
>
> **Response:** We appreciate the reviewer's feedback and acknowledge the importance of Pascal VOC as a benchmark in the graph matching community. However, previous research [2] has highlighted that Pascal VOC contains annotation errors, which can impact performance evaluations. While our method underperforms on Pascal VOC compared to some baselines, we have updated the main content to mention this limitation. The results related to Pascal VOC keypoint matching remain in the appendix due to space constraints.
>
> That said, our method demonstrates strong performance on other benchmarks, surpassing the state of the art in most cases. IMCPT 100 is the largest visual graph matching dataset and IMCPT datasets (IMCPT 50 and IMCPT 100) have the highest partial rate. Moreover, SPair-71K is another high-quality image keypoint matching dataset [2].
>
>
> [2] M. Rol´ınek, P. Swoboda, D. Zietlow, A. Paulus, V. Musil, and G. Martius, “Deep graph matching via blackbox differentiation of combinatorial solvers,” in European Conference on Computer Vision. Springer, 2020, pp. 407–424.
>
> Note : Please note that owing to the limitations on characters, we have provided our responses to the remaining questions of the reviewer as replies to this comment.

---

> ### Author Response · Authors · 2024-11-27
> **Responses for Reviewer aXCT's Comments**
>
> Please note that we have revised our response to comment W1 from the reviewer. Following this revision, we have provided our previous answers to comments W4, Q1, Q2, and Q3 as a separate comment here. We apologize for any inconvenience this may have caused.
>
> **W4:** *The paper omits two critical references [1,2], both of which have achieved better results on PascalVOC.*
>
> [1] Nurlanov et al., "Universe Points Representation Learning for Partial Multi-Graph Matching"
>
> [2] Jiang et al., "Learning Universe Model for Partial Matching Networks over Multiple Graphs"
>
> **Response:** We thank the reviewer for pointing out [1] and [2], which are indeed related to our work. In [1], the matching is performed at the class level of each image keypoint dataset, requiring the creation of a universe graph for each class. This approach relies on class-specific knowledge, such as number of distinct keypoints in each class, and assumes that the class of the two images (later converted to graphs) is known during both the training and inference stages. A key drawback of this approach is its inability to generalize to new, unseen classes (classes that are not in the training set) or to match random graphs that do not belong to a predefined class. Similarly, [2] also has the same limitations as it learns a sub universe for each class in training data, limiting its ability to generalize into images (graphs are later created using these images) from unseen classes during training.
>
> In contrast, our work focuses on the more general partial graph matching problem, where no additional information, such as class or the number of possible distinct keypoints, is assumed for the given graphs. This makes our approach applicable to datasets like IMCPT 50 and IMCPT 100, where training and test sets have disjoint classes. For example, IMCPT 100 contains 19 classes, 16 of which are used for training, with the remaining 3 reserved for testing. By not relying on class-specific information, our method can generalize to new scenarios, unlike the approaches in [1,2].
>
> We have added a discussion of these points in the related work section of the revised version to clarify the distinctions.
>
> **Q1:** *How effective is the applied bias term in identifying non-matching points, and what are the false positive and negative rates? It would be important to quantify the effectiveness of the proposed scheme.*
>
> **Response:**
>
> We thank the reviewer for providing us the opportunity to clarify on matching bias. The matching bias values ($\alpha$ and $\beta$) are learnt from data and have an influence on the final matching, but they alone do not explicitly identify non matching points or matching points. The effectiveness of learning $\alpha$ and $\beta$ can be observed by comparing the performance of the OPGM-rs model with the OPGM model (Table 1, Table 2 left)., where OPGM model assigns unit matching bias for each node (i.e., without learning matching bias).
>
> **Q2:** *What would the performance of other methods look like if they simply applied a thresholding scheme on their cost matrix similar to this study? An ablation study would be appreciated.*
>
> **Response:**
> We would like to highlight that our method differs fundamentally from a simple thresholding scheme. In our work, we show that the optimal partial transport problem in Eq. (6) can be reformulated and solved as a linear sum assignment problem in Eq. (9). The feasibility threshold is an inherent property derived from the optimization objective (Eq. (6)) itself, rather than being a manually set hyperparameter. Thus, our method *does not impose any thresholding on the cost matrix.* To avoid the confusion, we have revised the term "feasibility threshold" to "feasibility condition" in the revised manuscript.
>
> **Q3:** *How is the parameter $\rho$ chosen? Is it chosen by some heuristics? Fig. 2 (middle) suggests that  values supporting better performance are quite narrow (0.35-0.4), and these values vary across different tasks. Determining this parameter is one of the most challenging aspects of partial matching, so it is essential to understand how to set  without extensive trial and error.*
>
>
> **Response:** In our experiments, we performed a grid search over the range \([0.1, 0.5]\) with a step size of 0.1 to determine the best $\rho$ for each task. As the reviewer correctly pointed out, Fig. 2 (middle) demonstrates that the optimal range of $\rho$ is narrow. This behavior reflects the inherent sensitivity of partial matching to the unbalancedness parameter $\rho$. Developing heuristic-based or automated approaches for selecting $\rho$ remains a promising direction for future work.

---

> > ### Comment · Reviewer_aXCT · 2024-11-28
> >
> > Thank you for your detailed response. While your clarifications have addressed some concerns, there remain two critical points that need further discussion:
> >
> > * Regarding the claimed generalization capability: Your response implies a distinction from [1,2] in terms of not requiring class-specific information. However, there are several aspects that need to be addressed:
> >     * The dependency on pre-defined $\rho$ and learned task-specific $\alpha,\beta$ parameters appears to contradict the claim of a fully generalizable solver. These parameters effectively encode dataset-specific information during training, similar to how [1,2] encode class-specific information.
> >     * The evidence presented for generalization capability relies on the IMCPT dataset. However, this dataset is composed entirely of building images. Their classes is defined by which building is in the image. This doesn’t provide support for broad generalization claims. The gap between training and unseen classes in IMCPT is relatively narrow, as one can argue that all images are falling into one “building” category. A more compelling demonstration would involve generalization across more diverse semantic categories.
> >     * Given that $\rho$ is determined through grid search, it would be valuable to also include the computational cost of this parameter search compared to other methods.
> >
> > * Regarding the relationship to threshold-based approaches: Your response emphasizes that the method derives from optimization objectives rather than applying explicit thresholds. However, I would appreciate a more detailed technical discussion comparing your approach to existing learnable Sinkhorn implementations in GeoTransformer (specifically referencing their learnable $\alpha$ parameter as shown here https://github.com/qinzheng93/GeoTransformer/blob/e7a135af4c318ff3b8d7f6c963df094d7e4ea540/geotransformer/modules/sinkhorn/learnable_sinkhorn.py#L10). While the mathematical intuition might differs, the practical effect appears similar - both approaches learn parameters that determine which matching pairs to exclude.
> >
> >     I hope the author can clarify the fundamental operational differences between your feasibility condition and GeoTransformer's learnable Sinkhorn implementation and discuss any advantage in the proposed method. Now might pass the time you can revise your paper, but any empirical evidence would also be appreciated to demonstrage how these differences perform in practice.
> >
> >   These points are very important to support the novelty claim in this paper.

---

> ### Author Response · Authors · 2024-11-30
> **Fundamental difference between our method and GeoTransformer's learnable Sinkhorn**
>
> >Q1. Our method vs GeoTransformer's learnable Sinkhorn implementation
>
> We appreciate the reviewer’s feedback and the opportunity to clarify the novelty of our work. We emphasize that the learnable Sinkhorn implementation in GeoTransfer [1] and the optimization objective in Eq. (6) we propose address two fundamentally distinct problems.
>
> - **GeoTransfer [1]:** The learnable Sinkhorn implementation of [1] outputs a dense soft (non-binary, continuous) assignment matrix (confidence matrix) between two node sets [1]. Notably, the Sinkhorn method itself doesn't determine which matching pairs to exclude. This is because solving the entropy-regularized optimal transport problem [2] using Sinkhorn method always results in a soft assignment matrix where each entry is strictly positive (nonzero).  After obtaining this soft assignment using its Sinkhorn layer, [1] applies a mutual top-K mechanism, where a point match is selected if it is among the "k" largest entries of both the row and the column of the soft assignment matrix output by the Sinkhorn layer. As further information, we would like to mention that this overall process (described in Section 3.3 of [1]) will then result in a global dense point correspondences [1] , which is also different from solving partial graph matching problem and such dense correspondences do not adhere to the partial matching constraints considered in partial graph matching.
>
> - **Our work:**
>   The optimization objective proposed in Eq. (6), a key contribution of our work, addresses a distinct problem. That is, *given a soft assignment matrix between two nodes, how to derive a discrete one-to-one matching (exact partial assignment) adhering to partial matching constraints*. As stated in Theorem 6.2 of our work, solving the linear programming problem in Eq.(6) directly yields an exact partial assignment. More specifically:
>
>   - In the standard graph matching pipeline used by our OPGM method as well as in the baselines like AFAIT and GCAN-ILP, a Sinkhorn layer is used to derive a soft assignment matrix $S$, discussed in Section 7. $S$ is not the final solution for the partial graph matching problem. Our contribution is on providing a new mechanism to derive an exact partial assignment while preserving partial matching constraints, given the soft assignment matrix $S$ (We use $S$ to derive $C$ and then solve the optimal partial transport problem given in Eq.(6) to obtain an exact partial assignment).
>
> Note that, deriving an exact partial assignment from a (partial) soft assignment, such as the output of the learnable Sinkhorn layer, is a non-trivial task.
>
> - Thresholding mechanism won't work: A thresholding mechanism on a partial soft assignment matrix would fail because multiple values in a row or column can meet the threshold, violating the one-to-one constraint.
>
> - Matching with the highest valued entry won't work: Let us consider any node $ i \in V_S$ in the source graph, where the soft assignment scores with nodes from the target graph are represented by the $i^{\text{th}}$ row of the soft assignment matrix. Let $ j \in V_T$  be the node with which $ i $ has the highest confidence. However, another node from $V_S $ could already be matched with $ j$, resulting in multiple source nodes being matched to the same target node, there by violating the matching constraints.
>
> - Conventional assignment/matching algorithms such as the Hungarian algorithm won't work: These algorithms will always return a total injective mapping from source graph to the target graph.
>
> - k-assignment algorithms cannot be directly applicable: We don't know the number of matchings $k$ that should happen between the two graphs. In order to apply k-assignment algorithms, it is first needed to determine "k", which is a non-trivial task.
>
> While highlighting this fundamental difference, we provide a concise explanation of how the learnable $\alpha$ parameter in the Sinkhorn implementation of GeoTransfer[1] is fundamentally different from the optimization approach proposed in our work.
>
> [1] Qin, H. Yu, C. Wang, Y. Guo, Y. Peng, and K. Xu, “Geometric transformer for fast and robust point cloud registration,” in CVPR, 2022.
>
> [2] Cuturi, “Sinkhorn distances: Lightspeed computation of optimal transport,” NeurIPS, 2013.
>
> Please note that owing to space limitations, explanation continue in next comment.

---

> ### Author Response · Authors · 2024-11-30
> **Further explanations on previous comment**
>
> **learnable $\alpha$ parameter in Sinkhorn Implementation of [1]**
> The Sinkhorn implementation in [1] introduces a learnable parameter $alpha$, referred to as the dustbin parameter. This parameter fills the mass corresponding to newly added row and column (representing dustbin nodes) in the initial matrix. After solving the entropic optimal transport problem using the Sinkhorn algorithm, the dummy elements are discarded, resulting in a dense confidence matrix corresponding to a partial *soft* assignment matrix $P \in [0,1]^{m \times n}$ satisfying $P\mathbf{1}_n = \mathbf{1}_m$ and $P^{\intercal}\mathbf{1}_m \leq \mathbf{1}_n$ (For further details, refer to [3], where a similar technique is used to obtain a partial *soft* assignment matrix $P$).
>
> In contrast, partial graph matching aims to derive an exact partial assignment matrix $M \in$ {0,1}$^{m \times n}$ such that $M\mathbf{1}_n = \mathbf{1}_m$ and $M^{\intercal}\mathbf{1}_m \leq \mathbf{1}_n$, where entries are binary (0 or 1). Notably, a matrix $M \in \mathcal{M}$ cannot be directly obtained by solving an entropic regularized optimal transport problem using the Sinkhorn method, as it always results in a matrix where all entries are non-zero.
>
> Furthermore, the concept of using dustbin nodes for graph matching has been adapted in [4], which we consider as a baseline in our experiments. As their objective is to derive an exact partial matching $M \in \mathcal{M}$, the problem is formulated as an Integer Linear Programming (ILP) problem (which cannot be directly solved using Sinkhorn as Sinkhorn is based on continuous optimization and entropy regularization, thus relaxing integer constraints). [4] use a branch-and-bound algorithm to solve this ILP problem, which does not guarantee polynomial worst-case time complexity.
>
> **$\alpha$, $\beta$ in our work**
>
> For a given graph pair with $m$ nodes in one graph (source graph) and $n$ nodes in the other graph (target graph), $\alpha \in \mathbb{R}^m_{\geq 0}$, $\beta \in \mathbb{R}^n_{\geq 0}$  are derived based on the cross-graph node to node affinities (matrix $A$) through the end to end learning network and are then taken as input to the optimization problem in Eq. (6). They represent matching biases: $\alpha_i$ corresponds to the bias for node $i \in V_S$, and $\beta_j$ corresponds to the bias for node $j \in V_T$. Here, $V_S$ and $V_T$  denote the set of nodes in source graph and target graph respectively. Specifically, for a given graph pair, cost matrix $C$ provides information related to assignment cost of each node pair while $\alpha$ and $\beta$ provides node level information of source graph and target graph, respectively.
>
> **Feasibility condition vs learnable $\alpha$ in Sinkhorn Implementation of [1]**
>
> For a given node $i \in V_S$ and a node $j \in V_T$, feasibility condition states the maximum cost that can exist between two nodes to make it feasible for $i$ and $j$ to be matched in an optimal solution of Eq. (6). If cost between two nodes ($C_{ij}$) exceeds feasibility condition,  $i$ and $j$ will never be matched in an optimal solution for Eq. (6) (Thoerem 5.1). However, it should be noted $C_{ij}$ being less than feasibility condition does not guarantee that $i$ and $j$ will be matched in an optimal solution for Eq. (6) (making this approach significantly different from thresholding). The final decision on whether $i$ and $j$ will be matched is determined w.r.t. to the optimality in Eq. (6) considering all other possible assignments (a combinatorial optimization problem). Therefore, feasibility condition is only a property of a solution to Eq.(6).
>
>  In contrast, $\alpha$ in GeoTransformers [1] is a learnable parameter of the Sinkhorn layer in [1] that is used to control the fillable mass for dustbin row and column discussed in that study.
>
> As described above, Sinkhorn implementation in [1] returns a soft assignment and does not solve the partial graph matching problem. In contrast, our method returns an exact partial assignment  adhering to partial matching constraints of the partial graph matching problem (while being mathematically optimal to the optimal partial transport based objective given in Eq. (6) ). This is the main advantage of our approach compared to the Sinkhorn implementation in [1] .
>
> [1]  Qin, H. Yu, C. Wang, Y. Guo, Y. Peng, and K. Xu, “Geometric transformer for fast and robust point cloud registration,” in CVPR, 2022.
>
> [2] Cuturi, “Sinkhorn distances: Lightspeed computation of optimal transport,” NeurIPS, 2013.
>
> [3] E. Sarlin, D. DeTone, T. Malisiewicz, and A. Rabinovich, “Superglue: Learning feature matching with graph neural networks,” in CVPR, 2020.
>
> [4] . Jiang, H. Rahmani, P. Angelov, S. Black, and B. M. Williams, “Graph-context attention networks for size-varied deep graph matching”, CVPR 2022.

---

> ### Author Response · Authors · 2024-11-30
> **Distinction between our method and universe point representaion learning based graph matching approaches**
>
> >Q2. Distinction between our method and universe point representaion learning based graph matching approaches [1] [2]
>
> [1] Nurlanov et al., "Universe Points Representation Learning for Partial Multi-Graph Matching"
>
> [2] Jiang et al., "Learning Universe Model for Partial Matching Networks over Multiple Graphs"
>
> **Generalization Capability: Distinction from [1] and [2]**
>
> **Response :** We appreciate the opportunity to clarify the distinction between our method and those in [1] and [2].
>
> While we do not claim our method to be a fully generalizable solver, our approach is a general graph matching solver applicable to open-set problems, unlike [1] and [2], which are restricted to closed-set scenarios.
>
> 1. Closed-Set Limitation of [1] and [2]: Methods in [1] assume the class of a given graph pair is known during the inference (require class labels during the inference phase) and learns a universe graph for each class in training set, and [2] is based on sub-universe graphs created for each class in the training set. Therefore, in both approaches, the universe graph creation is based on the assumption that the set of classes in training is identical to the set of classes in testing (A closed-set setting [3]). As both methods rely on matching based on universe graphs learned based on each specific class in the training set, they inherently restrict their use to seen classes during training and cannot be applied to match graphs belonging to unseen classes in the inference phase (as no universe graph has been learned considering unseen classes). A simple example to explain this would be a setting where an unseen class contains graphs that have more distinct nodes than the number of nodes in the universe graph that is learned during training phase (here, the universe graph based approach will not work). Thus, [1] [2] consider partial graph matching in a closed-set problem setting [3].
>
> 2. Open-Set Capability of Our Method: The matching mechanism in our method does not rely on learning universe graphs or similar structures from training data, which allows it to handle unseen classes and arbitrary graph structures (with varying sizes and structures) during inference.
>
> Thus, our method addresses a broader scope of the partial graph matching problem compared to the closed-set approach of [1] and [2].
>
> **Generalization Discussion on the IMCPT Dataset**
>
> We use the IMCPT dataset to highlight the limitations of [1] and [2] in handling unseen categories during inference. The IMCPT dataset's standard experimental setup [4] includes a test set with classes not present in the training set, representing an open-set setting. Since [1] and [2] require training data from each class to construct a universe graph, they cannot operate in this setup.
>
> While the IMCPT dataset primarily consists of building images, each class vary in shape, structure, and no. of distinct keypoints in each class (e.g., Colosseum vs. St. Peter’s Square). Our method, along with other general graph matching solvers like AFAT-I and AFAT-U, effectively handles open set experimental setup of IMCPT 100.
>
> It should be further noted that, the standard experimental setups of SPair-71K and Pascal VOC consider a closed-set experimental setting (Same set of classes in testing and training) [4]. This is the primary reason why we discussed on IMCPT 100 dataset, which distinguishes itself by having an open-set experimental setting.
>
>
>
> References
>
> [1] Nurlanov et al., "Universe Points Representation Learning for Partial Multi-Graph Matching"
>
> [2] Jiang et al., "Learning Universe Model for Partial Matching Networks over Multiple Graphs"
>
> [3]  Zheng, L. Zheng, Z. Hu, and Y. Yang, “Open set adversarial examples,” arXiv preprint arXiv:1809.02681, vol. 3, 2018
>
> [4] Runzhong Wang, Ziao Guo, Shaofei Jiang, Xiaokang Yang, and Junchi Yan, "Deep learning of partial graph matching via differentiable top-k." IEEE/CVF Conference on Computer Vision and Pattern Recognition 2023.

---

> ### Author Response · Authors · 2024-11-30
> **Determination of $\rho$ and Training Phase**
>
> **Determination of $\rho$**
>
> The determination of $\rho$ via grid search is not computationally exhaustive. A clear understanding of $\rho$'s optimal range can be obtained after observing results from the first training epoch. For example, the table below demonstrate results after first training epoch in IMCPT 100 dataset for different $\rho$ values, providing sufficient insight into the performance trends of each $\rho$ values, significantly reducing the search effort.
>
> | $\rho$                   | 0.1   | 0.2   | 0.3   | **0.4**   | 0.5   |
> |---------------------------|-------|-------|-------|-------|-------|
> | **Mean Matching F1-Score** | 40.45 | 55.28 | 65.03 | **69.92** | 67.92 |
>
> *Table: Result for IMCPT 100 after 1 training epoch under varying $\rho$ values.*
>
> **Computational complexity in inference phase**
>
> We clarify that hyperparameter $\rho$ is determined only during the training phase and remains fixed during inference (The $\rho$ value that was determined in the training stage is used in inference phase). Thus, $\rho$ does not introduce additional computational complexity in the inference phase.
>
> **Training Phase:**
>
> We have included a comparison of the training of our method (OPGM and OPGM-rs) with the closest competing baselines, AFAT-I and AFAT-U. Like other deep graph matching methods, our approach requires configuring standard hyperparameters such as learning rate, epochs, and batch size. Additionally, OPGM uses one specific hyperparameter (\(\rho\)), while OPGM-rs uses two (\(\rho\), \(\lambda\)).
>
> In contrast, AFAT-I and AFAT-U solve partial graph matching as a k-assignment problem, requiring separate learnable modules to determine the "k" number of node pairs to be matched. They also have 4 additional hyperparameters that needs to be configured for each dataset (see hyperparameters under AFA: section of the configuration file https://github.com/Thinklab-SJTU/ThinkMatch/blob/master/experiments/gcan-afat/vgg16_gcan-afat-i_imcpt_100_stage1.yaml.
> Furthermore, AFAT-I and AFAT-U has 3 training stages, where each training stage has a different set of hyperparameters (the hyperparameters such as learning rate are also varied across each training stage) https://github.com/Thinklab-SJTU/ThinkMatch/tree/master/experiments/gcan-afat.
>
> Thus, our method requires significantly fewer hyperparameters during training compared to AFAT-I and AFAT-U. However, since the number of hyperparameters and their respective search spaces vary across methods, a fair comparison of hyperparameter tuning is challenging. Notably, our models OPGM and OPGM-rs outperform AFAT-I and AFAT-U on IMCPT 100 and Spair-71K datasets, while showing comparable performance on IMCPT 50 dataset.
>
> Moreover, we provide a comparison of training times for our best-performing model, OPGM-rs, against the closest competing methods, AFAT-I and AFAT-U, considering the IMCPT 100 dataset.
>
> | **Metric**                                                | **OPGM-rs** | **AFAT-I** | **AFAT-U** |
> |-----------------------------------------------------------|-------------|------------|------------|
> | **No. of Training Stages**                                | 1           | 3          | 3          |
> | **Batch Size**                                            | 4           | 4          | 4          |
> | **Total No. of Training Epochs**                          | 25          | 35         | 30         |
> | **Total Number of Training Samples (graph pairs) per epoch** | 4000        | 4000       | 8000       |
> | **Total Number of Training Samples (graph pairs)**       | 4000*25     | 4000*35    | 8000*30    |
> | **Total Training Time**                                   | 5h 29m 53s  | 16h 01m 10s| 26h 43m 26s|
> | **Average number of Training Samples Processed per second** | 5.05        | 2.43       | 2.49       |
>
> *Table: Method comparison according to training time.*
>
> It is worth noting that the total training time for our method is significantly lower, primarily due to a higher number of samples processed per second in our method and fewer number of training epochs. AFAT-I and AFAT-U have added complexities due to the requirement of learning neural modules to estimate $k$ , making the training process slower compared to our method.

---

> > ### Comment · Reviewer_aXCT · 2024-12-03
> >
> > Thank you for your detailed response. While I appreciate your thorough explanation, I have remaining concerns about the method's fundamental novelty.
> >
> > Your response describes solving Eq(6) to derive partial assignments for node pairs satisfying the feasibility condition. However, examining the solution in Section 6, I find the practical mechanism is mathematically equivalent to appending dummy nodes with affinity values $\rho(\alpha+\beta)$ to the affinity matrix before applying standard assignment algorithms - a formulation virtually identical to appending dummy nodes with learnable affinity $\alpha$ before applying Sinkhorn/Hungarian algorithms (as implemented in GeoTransformer’s learnable Sinkhorn). Given these similarities, I believe the paper's primary contribution lies in establishing theoretical connections between partial matching and optimal transport, rather than introducing novel methodology.
> >
> > Regarding the generalization ability claim, the “open-set ability” is from the neural network's learned feature representations rather than the proposed solver, whose parameters remain heavily dependent on the data distribution. While we may differ in perspective here, I would argue that truly “open-set” solvers align more closely with approaches presented in RRWM[3] and ZAC[4].
> >
> > Given these considerations, I maintain my current rating.
> >
> > [3] Reweighted Random Walks for Graph Matching, ECCV 2010
> > [4] Zero-Assignment Constraint for Graph Matching with Outliers, CVPR 2020

---

> ### Author Response · Authors · 2024-12-03
>
> We kindly request to ignore this comment (by authors).
>
> Our **responses for reviewer's new comments** can be found below as replies.

---

> ### Author Response · Authors · 2024-12-03
> **Response (1) by Authors for Reviewer's new comments**
>
> >**Q1** *I believe the paper's primary contribution lies in establishing theoretical connections between partial matching and optimal transport, rather than introducing novel methodology.*
>
> **Response:** We respectfully disagree with the reviewer on this. We infact propose a novel methodology as described below. In this work, we consider partial graph matching in the lense of optimal partial transport (not optimal transport). However, solving a general optimal partial transport problem will typically result in a transportation plan that does not provide an exact partial matching/exact partial assignment between the sets.
>
> Our **first contribution** is proposing a optimal partial transport based optimization objective (eq.(6)) that guarantees to have an optimal solution inducing an *exact partial matching* (Theorem 5.2). The optimization intuition behind our method is similar to optimal partial transport. Within our optimization objective given in Eq. (6), there is a cost for two nodes being matched (as indicated by term $\langle \pi, C \rangle_F$ in Eq. (7)) and there is a cost for leaving a node unmatched (as indicated by weighted total variation (divergence function) term $\rho\left( \langle \alpha, \mathbf{1}_m - \pi_1 \rangle
>     + \langle \beta, \mathbf{1}_n - \pi_2 \rangle \right)$ in Eq.(7)). Therefore, our proposed mechanisms achieves a balance between matchings and non matchings, resulting in an partial matching that is optimal w.r.t. Eq. (6). Thus, solving Eq.(6) which is based on optimal partial transport will return the partial matching between two given graphs.
>
>
> However, solving the optimization objective given in Eq.(6) efficiently and finding the optimal solution inducing a one to one matching constraint is non trivial.
>
> In order to solve Eq. (6) efficiently, we further explore the solution space of Eq. (6) and theoretical properties of Eq. (6) ( feasibility condition is one such thereotical property) and was able to establish a Theoretical connection between Eq. (6) and the linear sum assignment problem in Eq. (9). This is a significant result (**second contribution**), as it enables the solving the partial graph matching problem we consider in Eq. (6) within a cubic worst case time complexity.
>
> Then as our **third contribution**, we introduce a deep graph matching architecture that utilizes these theoretical results to solve partial graph matching.
>
> > **Q2**  *I find the practical mechanism is mathematically equivalent to appending dummy nodes with affinity values $\rho(\alpha+\beta)$  to the affinity matrix before applying standard assignment algorithms. A formulation virtually identical to appending dummy nodes with learnable affinity $\alpha$ before applying Sinkhorn/Hungarian algorithms*
>
> **Response:** The mechanism we propose to solve Eq. (6) is not mathematically equivalent to assigning affinity values for dummy nodes in an affinity matrix and then solving an standard assignment problem.
>
> Explanation : Within our mechanism to solve Eq.(6), we have a mapping from $C \in \mathbb{R}^{m \times n}$ to $\overline{C} \in \mathbb{R}^{n \times n}$ as given in Eq.(8), where we assign affinity of $\rho(\alpha_{*}+\beta_{j})$ for dummy elements added (that corresponds to $i > m$). However, it is worth noting that **we assign new cost values considering non-dummy nodes** (actual nodes) as well (where $i \leq m$ in Eq. (8)).
> These cost modifications in Eq.(8) for non-dummy nodes (actual nodes) are necessary to preserve the theoretical connection between an optimal solution of Eq. (9) to an optimal solution of Eq. (6). This mapping given in Eq. (8) is not possible without the theoretical result we derive in Theorem 5.1 (which is on the feasibility condition). Therefore, we would like to emphasize that feasibility condition is not a random threshold value, but it is a theoretical property of an optimal solution of Eq. (6). Thus, result of Theorem 5.1 plays a significant role in establishing the theoretical connections between optimal solution of Eq. (9) and Eq. (6).
>
> Moreover, we also uses a **closed-form mapping** given in Eq. (10) to derive an optimal solution for Eq.(6) from an optimal solution of Eq. (9) (because solving Eq. (9) using the Hungarian algorithm itself will not result in an optimal solution for Eq. (6)).
>
> Therefore, the overall mechanism we propose to solve Eq. (6) is not mathematically equivalent to assigning affinity values for dummy nodes in the affinity matrix and then solving an standard assignment problem. Our cost modifications in Eq.(8) are not limited for dummy elements and we also uses the closed form mapping in Eq. (10) to obtain the solution for Eq. (6).
>
> Further, establishing the theoretical connection between the optimal partial transport objective in Eq. (6) and the linear sum assignment problem given in Eq. (9) is a non-trivial task. We use the theoretical results of Theorem 5.1,Theorem 5.2, Theorem 6.1, and Theorem 6.2 to establish the connection.
>
> .

---

> ### Author Response · Authors · 2024-12-03
> **Response (2) by Authors for Reviewer's new comment**
>
> >**Q3** :Regarding the generalization ability claim :
>
> When distinguishing our method from [1] and [2] in our previous response titled "*Distinction between our method and universe point representaion learning based graph matching approaches*", what we emphasized was that while [1] and [2] consider partial graph matching in a closed-set setting [3], our method, along with baselines such as AFAT and GCAN-ILP, addresses partial graph matching in an open-set setting [3]. This approach makes our method applicable to open-set experimental setups (such as IMCPT) and capable of working with graphs of varying sizes and structures.
>
> The limitations in [1] and [2] stem from their approach of learning universe graphs based solely on classes in the training set, assuming the training and test class sets are identical (a closed-set assumption [3]). A simple example illustrates this limitation of closed-set assumption: since the universe graph has a predefined size based on the training data classes, if two graphs from an unseen class are presented with a number of distinct nodes in those two graphs exceeding the number of nodes in the universe graph, the universe graph will fail.
>
> [1] Nurlanov et al., "Universe Points Representation Learning for Partial Multi-Graph Matching"
>
> [2] Jiang et al., "Learning Universe Model for Partial Matching Networks over Multiple Graphs"
>
> [3] Zheng, L. Zheng, Z. Hu, and Y. Yang, “Open set adversarial examples,” arXiv preprint arXiv:1809.02681, vol. 3, 2018

---

> ### Author Response · Authors · 2024-12-04
> **Response (3) by Authors for Reviewer's new comment (Summary)**
>
> In this response, we provided a summary addressing reviewer's concern on novelty of our work in addition to the detailed explanation given in our previous comment titled "Response (1) by Authors for Reviewer's new comments".
>
> >**Reviewer's Comment :** *I find the practical mechanism is mathematically equivalent to appending dummy nodes with affinity values $\rho(\alpha+\beta)$ to the affinity matrix before applying standard assignment algorithms*
>
> We respectfully disagree with this. As described below, the mechanism we propose to solve Eq.(6) is not mathematically equivalent to assigning  affinity values for dummy nodes in an affinity matrix and then solving an standard assignment problem.
>
> Explanation : Eq.(6) is the optimal partial transport-based optimization objective we propose to solve the partial graph matching problem. Solving Eq.(6) will return a partial graph matching solution. Within our work, we propose a novel mechanism to exactly solve this optimization objective (Eq.(6)) efficiently. Within this mechanism we have a mapping from $C$ to $\overline{C}$ in Eq.(8), where the cost modifications are not limited to dummy nodes, but are done considering **non-dummy nodes** (actual nodes) as well. The way the cost is assigned considering non-dummy nodes (where $1 \leq i \leq m$ in Eq. (8)) is different from the way the cost is assigned considering dummy nodes (where $m < i \leq n$ in Eq. (8)). **These cost modifications considering non-dummy nodes are essential to preserve theoretical connection between an optimal solution for Eq. (6) and an optimal solution for Eq. (9)**. The cost modifications (Eq.(8)) as well as the **closed-form mapping** we provide in Eq. (10) are based on the theoretical results of Theorem 5.1, Theorem 5.2, Theorem 6.1, and Theorem 6.2.
>
> Therefore, the above mentioned overall mechanism we propose is necessary to exactly solve Eq.(6) efficiently by exploiting the theoretical connection it has with the linear sum assignment problem given in Eq. (9).
>
> Because of the above mentioned reasons, our mechanism is not mathematically equivalent to adding dummy nodes with affinity values to the affinity matrix and then solving a standard assignment problem. **More specifically, our mechanism uses a mapping (Eq.(8)) that assigns cost for each possible assignment in $\overline{C}$ in Eq. (8) (considering both non-dummy nodes and dummy nodes)  and also uses a closed-form mapping in Eq. (10) to obtain a solution for Eq. (6) from a solution of Eq. (9), thus making our proposed mechanism fundamentally different from just adding dummy nodes with affinity values to the affinity matrix and then solving a standard assignment problem**.
>
>
> **Difference Between Our Mechanism and GeoTransfer's Mechanism**
>
> As described above, just adding dummy nodes with affinity values to the affinity matrix before applying standard assignment algorithms will not solve the partial graph matching problem we consider in Eq. (6). This alone makes our approach different from the approach discussed in the learnable Sinkhorn layer of GeoTransformer [1], which assigns learnable affinity for dustbin (dummy) column and dustbin row , and discards the dustbin column (last column)  and dustbin row (last row) later after solving an entropy regularized optimal transport problem.
>
> Moreover, another fundamental difference is that GeoTransformer with its Sinkhorn implementation is not a solver for the partial graph matching problem (as detailed in our previous comment titled "Fundamental difference between our method and GeoTransformer's learnable Sinkhorn").
>
>
> **Summary of Contributions and Novelty**
>
> In our work, we propose a novel optimization objective based on optimal *partial* transport to solve the partial graph matching problem. Next, we propose a novel mechanism to exactly solve the proposed optimization objective efficiently with theoretical guarantees. We then propose how these theoretical results can be integrated into an end-to-end learning setting to solve the partial graph matching problem.
>
> [1] Qin, H. Yu, C. Wang, Y. Guo, Y. Peng, and K. Xu, “Geometric transformer for fast and robust point cloud registration,” in CVPR, 2022.

---

### Official Review · Reviewer_HfMe · 2024-10-31

**Soundness:** 3
**Presentation:** 2
**Contribution:** 3
**Rating:** 6
**Confidence:** 3

**Summary:**

The paper proposes a new objective for partial graph matching based on total variation and partial optimal transport. The paper shows that this formulation leads to a specialized version of optimal transport, and one optimal solution exists in the unit-weight setting.
Then, the paper proposes to solve the problem by adapting the Hungarian algorithm and training a neural network to learn two components: the matching cost matrix and the biases to represent the matched and unmatched nodes.

The experiments cover protein-protein interaction networks and image key-point matching, showing results under different noise levels and analyzing the method's efficiency.

**Strengths:**

1) The paper addresses a challenging and fundamental problem, with an impact on several down stream domains. The derivation lead to a formulation includes also a computational complexity statement for the problem, which seems useful for future works.
2) The general formulation is interesting and leads to a concrete network formulation. The network seems practical, and the authors have attached the code, so I am confident the work can be reproduced.
3) Although the performance does not push the state of the art further, the method seems particularly more computationally efficient.

**Weaknesses:**

1) The method requires setting a series of hyperparameters tailored for the domain, which might be difficult in practice. For example, by my understanding, the parameter \rho seems quite important and encodes the amount of partiality. This is quite an assumption, and it would be nice to find some heuristics to fix it automatically (e.g., the ratio of the number of nodes or the graph radius, etc.). Could you offer a comment on this, and in case a possible heuristic? Is it possible to explicitly relate \rho to the amount of partiality?
2) Since this work is a bit outside my direct expertise, I had a hard time grasping all the details, especially building an intuition. The paper itself does not show visual results, and then it is not straightforward to understand how concretely the proposed methodology differentiates in terms of artifacts (e.g., is the error caused by wrong matching bias and partiality identification, or more about mismatching?) would be interesting to add some qualitative examples, especially for keypoint, that is particularly easy to interpret. Did you observe any specific failure mode or particularly challenging setting? Could you provide a visualization to support the intuition about the main source of the error?

**Questions:**

Please, refer to the weaknesses section comments for the major questions I would see addressed in the rebuttal.

Minor:
1) line 075: multiple missing spaces: "transportidentifies", "transportcan", "transportdoes"
2) The recent work "Spectral Maps for Learning on Subgraphs", Pegoraro et al., NeuReps 2023, seems relevant for the paper. In particular, the paper suggest to use spectral properties to infer properties on subgraphs, and also correspondence. Similarly "A functional representation for graph matching", Wang et al., PAMI 2019, propose to use spectral properties to compute matching on (full) graphs. I believe a discussion on this line of works would be interesting.
3) How does the method perform in case of rewiring of the partiality?

---

> ### Author Response · Authors · 2024-11-25
> **Responses for Reviewer HfMe's Comments**
>
> **W1:** *The method requires setting a series of hyperparameters tailored for the domain, which might be difficult in practice. For example, by my understanding, the parameter $\rho$ seems quite important and encodes the amount of partiality. This is quite an assumption, and it would be nice to find some heuristics to fix it automatically (e.g., the ratio of the number of nodes or the graph radius, etc.). Could you offer a comment on this, and in case a possible heuristic? Is it possible to explicitly relate $\rho$ to the amount of partiality?*
>
>
>
>
> **Response:** Thank you for the insightful comment. We agree that the choice of the hyperparameter $\rho$, which encodes the amount of partiality, is important and can influence the performance of the method. However, our method requires fewer hyperparameters overall, as the cost matrix $C$ and matching biases $\alpha,\beta$ are not hyperparameters. Instead, they are learned directly from data.
>
> To address the selection of $\rho$, one heuristic could involve analyzing the distribution of graph sizes (number of nodes) in the dataset. In supervised settings with ground truth, the average partiality ratio between graph pairs in a sample can provide a baseline for determining $\rho$. This approach ensures $\rho$ is data-driven while remaining flexible for refinement based on domain knowledge or validation performance.
>
> Empirically, we observe that $\rho$ modulates the trade-off between matching accuracy and allowable mismatches, effectively controlling the degree of partiality. Future work could explore domain-specific properties or self-supervised techniques to further automate and optimize $\rho$ selection.
>
> **W2:** *Since this work is a bit outside my direct expertise, I had a hard time grasping all the details, especially building an intuition. The paper itself does not show visual results, and then it is not straightforward to understand how concretely the proposed methodology differentiates in terms of artifacts (e.g., is the error caused by wrong matching bias and partiality identification, or more about mismatching?) would be interesting to add some qualitative examples, especially for keypoint, that is particularly easy to interpret. Did you observe any specific failure mode or particularly challenging setting? Could you provide a visualization to support the intuition about the main source of the error?*
>
>
> **Response:**  The feasibility of matching a pair of nodes $i$ and $j$ is influenced by their cost $C_{ij}$, the matching biases $\alpha_i$ and $\beta_j$, and the hyperparameter $\rho$. A specific failure mode we observed is when node pairs that should be matched according to the ground truth are deemed infeasible by the optimization objective in Eq. (9) (as described in Theorem 5.1). Consequently, these nodes remain unmatched in the final solution. This failure mode typically arises when the graph matching network ${NN}_\theta$ assigns an excessively low affinity (resulting in a high matching cost) to pairs of nodes that should otherwise be matched.
>
> We quantify the error by analyzing:
>
> 1. Partiality error: The percentage of true matches (as per the ground truth) that fail to meet the feasibility condition outlined in Theorem 5.1. This error reflects mismatches caused by incorrect partiality identification involving matching biases.
>
> 2. Mismatching error: The percentage of true matches that are not identified even though they meet the feasibility condition.
>
>
> Below, we present the error rates for Partiality Identification and Mismatching across various datasets, and a visualization of the results is provided in the appendix of the revised manuscript.
>
> | **Dataset**  | **Partiality  (%)** | **Mismatching (%)** | **Total (%)** |
> |--------------|-----------------------------------:|---------------------:|--------------:|
> | IMCPT 100    |                         21.59    |               4.97  |          26.56 |
> | IMCPT 50     |                         20.15    |               6.12  |          26.27 |
> | SPair-71K    |                         29.07    |               9.64  |          38.71 |
> | Pascal VOC Keypoints         |                         22.50    |              13.28 |          35.78 |
>
> **Q1:** *line 075: multiple missing spaces: "transportidentifies", "transportcan", "transportdoes"*
>
> **Response:** We thank the reviewer for pointing this out. We have fixed this in the revised manuscript.
>
> Note : Please note that owing to the limitations on characters, we have provided our responses to the remaining questions of the reviewer as replies to this comment.

---

> ### Author Response · Authors · 2024-11-25
> **Responses for Reviewer HfMe's Comments**
>
> **Q2:** The recent work "Spectral Maps for Learning on Subgraphs", seems relevant for the paper. In particular, the paper suggest to use spectral properties to infer properties on subgraphs, and also correspondence. Similarly "A functional representation for graph matching", propose to use spectral properties to compute matching on (full) graphs. I believe a discussion on this line of works would be interesting.
>
> **Response:** Thank you for pointing out these relevant works. We have added a discussion on them in the revised manuscript.
>
> "Spectral Maps for Learning on Subgraphs" uses spectral representations to compactly encode mappings between graphs and subgraphs within graph learning pipelines. However, the key distinction between their approach and ours is that they do not address or guarantee the assignment of nodes between the node sets of two graphs in accordance with partial matching constraints, which is the primary focus of our work.
>
> On the other hand, "A Functional Representation for Graph Matching" addresses the graph matching problem by leveraging spectral properties but focuses exclusively on graph matching. It does not tackle the challenges associated with partial graph matching, which is the core problem we address.
>
> **Q3:** *How does the method perform in case of rewiring of the partiality?*
>
> **Response:**
>
> Thank you for the question. If by "rewiring of partiality," the reviewer refers to evaluating the method's performance under different partiality ratios between graph pairs, this evaluation poses a challenge in our context. The partial graph matching datasets we use have predefined ground truth mappings with fixed partiality ratios for a given graph pair, which limits the ability to alter or rewire partiality without compromising the integrity of the ground truth. These datasets are specifically constructed to represent partial correspondences in real world data (eg : images), and modifying the partiality ratios would undermine the reliability of these predefined mappings.
>
> However, we believe that future work could explore generating synthetic datasets with adjustable partiality ratios to better evaluate the robustness of our method under varying levels of partial correspondence.

---

> > ### Comment · Reviewer_HfMe · 2024-11-26
> > **Post-rebuttal**
> >
> > I thank the authors for their reply to my concerns. I appreciate their efforts, especially in providing further experiments and insight into the method failure modes. I have a further question: what can be the cause of $NN_\theta$ assigning too low affinity?
> >
> > I see the updated title, but I believe it is too generic. I believe that the comment from Reviewer aXCT was about having a more specific title rather than a more general one.
> >
> > About "rewiring of partiality": I meant to consider the case in which the partial graph has a similar ratio to the one observed in the training set, but it faces a change in local edge connections. Real-world cases for this are the opening\closing of roads\rails that connect cities or the evolution of slime mold.
> >
> > Best.

---

> ### Author Response · Authors · 2024-11-27
> **Responses for Reviewer HfMe's Post-Rebuttal Comments**
>
> We thank the reviewer for their post-rebuttal comments and suggestions.
>
> > What can be the cause of $NN_{\theta}$ assigning too low affinity?
>
> In our experiments, for a fair comparison with other works, we use the graph matching network architecture GCAN [1] as $NN_{\theta}$. There are several possible causes of $NN_{\theta}$ assigning too low affinity for some node pairs that should be matched:
>
> - *Limitations in Generalizing to Unseen Data:* In the partial graph matching datasets considered, the training and testing sets are disjoint. Specifically, in datasets IMCPT 50 and IMCPT 100, training is performed on images from 16 out of 19 available classes, while testing is conducted on the remaining 3 classes. This class separation introduces significant generalization challenges, as the model must infer similarities in node representations for entirely unseen class configurations. Consequently, $NN_{\theta}$ may struggle to correctly identify some corresponding/similar points across these novel class instances, potentially underestimating node affinities.
>
> - *Challenges due to structural properties:* Moreover, as we consider the partial graph matching datasets, two graphs can have different number of key points annotated even if they belong to the same class. Thus, the corresponding nodes (nodes that should be matched) might have different number of neighbors, thus different connectivity. These structural variations can make it difficult for $NN_{\theta}$ to assign high affinity for corresponding nodes. Moreover, certain classes in visual graph matching datasets (such as "chair", "table", and "sofa") can present inherent structural challenges. These objects can often exhibit symmetrical properties that can confuse the matching process. For instance, keypoints related to right leg of a chair might be misinterpreted as keypoints related to left leg.
>
> - *Poor and Ambiguous Annotations:* Annotation errors can also impact the model's learning process. This is particularly evident when observing the results related to classes like *table* and *sofa* of Pascal VOC keypoint matching (Table 3 in appendix), which are known to have poor and ambiguous annotations [2]. Such annotation inconsistencies can lead $NN_{\theta}$ to learn incorrect or misleading representations that compromise its ability to accurately match corresponding nodes.
>
> > I see the updated title, but I believe it is too generic. I believe that the comment from Reviewer aXCT was about having a more specific title rather than a more general one.
>
> We appreciate reviewer's suggestion and we agree with the reviewer. We have changed the title of the paper to "A Novel Optimization Framework for Partial Graph Matching via Optimal Partial Transport". If the reviewer has any further suggestions regarding the title, we would appreciate their feedback.
>
> > About "rewiring of partiality": I meant to consider the case in which the partial graph has a similar ratio to the one observed in the training set, but it faces a change in local edge connections. Real-world cases for this are the opening\closing of roads\rails that connect cities or the evolution of slime mold.
>
> We thank the reviewer for the clarification.
>
> In our partial graph matching experiments, each graph pair (in both training and testing) is different, with both source and target graphs changing for each instance. Consequently, the partiality ratio between two graphs as well as the connectivity structure (local edge connections) of both source and target graphs vary across each instance (in both training and testing phases). Both source and target graphs can contain nodes that will not be matched, and the size and connectivity structures of these graphs differ significantly. Furthermore, our testing and training sets are disjoint. Overall, this creates a more challenging setup than simply re-wiring partiality, as we do not assume the partiality ratio or structural connectivity (of both source and target graphs) remains fixed.
>
> Our experiments on Protein-Protein Interaction (PPI) network matching may provide insights into the performance of our method when connectivity structures or local edge connections undergo changes.  We match protein protein interaction (PPI) network of yeasts with with its modified versions, created by adding edges that corresponds to low-confidence protein-protein interactions (the percentage of modified edges change from 5\%-25\%). The results are provided in Figure 2 (left) and Table 4 (appendix).
>
> [1] Z. Jiang, H. Rahmani, P. Angelov, S. Black, and B. M. Williams, “Graph-context attention networks for size-varied deep graph matching,” in Proceedings of the IEEE/CVF Conference on Computer Vision and Pattern Recognition, 2022, pp.2343–2352.
>
> [2] M. Rol´ınek, P. Swoboda, D. Zietlow, A. Paulus, V. Musil, and G. Martius, “Deep graph matching via blackbox differentiation of combinatorial solvers,” in European Conference on Computer Vision. Springer, 2020, pp. 407–424.

---

> > ### Comment · Reviewer_HfMe · 2024-11-27
> > **About the title**
> >
> > Thank you for your further clarifications.
> >
> > A small note about the title: I agree that the new one is more communicative. Although, some wording looks a bit unnecessary (e.g., "Novel"). A simple suggestion could be "Learning Patrial Graph Matching via Partial Optimal Transport," but it is also a matter of personal taste.
> >
> > As of now, I do not have further questions, and I am looking forward to hearing other reviewers' opinions.

---

> ### Author Response · Authors · 2024-11-27
> **Thank you**
>
> Thank you for your feedback. We appreciate your input on the title. We’ve updated it to “Learning Partial Graph Matching via Optimal Partial Transport”.

---

### Official Review · Reviewer_i3qN · 2024-11-04

**Soundness:** 3
**Presentation:** 3
**Contribution:** 2
**Rating:** 6
**Confidence:** 4

**Summary:**

The paper proposes a new optimization framework for partial graph matching, a variant of traditional graph matching that allows for some nodes to remain unmatched. This flexibility accommodates complex real-world scenarios, where not all entities have a direct counterpart. The authors introduce an optimization objective that balances matched and unmatched nodes using concepts from optimal partial transport, providing an efficient, cubic-time complexity solution via the Hungarian algorithm.

**Strengths:**

- The paper applies concepts from optimal partial transport to graph matching, resulting in a framework that handles partial assignments rigorously. By connecting the problem to the linear sum assignment, the authors leverage the Hungarian algorithm, achieving cubic time complexity—an important advancement for handling larger datasets.

- Experimental results on multiple datasets illustrate the proposed method’s robustness and competitive edge over established baselines, particularly in terms of matching F1-scores and runtime efficiency.

- The end-to-end architecture, with its unique loss function, offers practical advantages for data-driven applications, adapting dynamically to different matching scenarios.

**Weaknesses:**

- The paper shows that the proposed model’s performance decreases dramatically with increasing noise level, with the PPI dataset. Although the proposed method performs better than other methods generally. The performance decrease rate is similar or enven larger than other methods, as shown in Fig 2 (left).
- It seems the approach’s performance is sensitive to the unbalancedness parameter ( \rho ), as shown in sensitivity analyses.
- For applications where annotations are inconsistent or sparse, such as biological or social network datasets, the model’s dependence on well-annotated data could limit its generalizability.

**Questions:**

please address the points mentioned as weaknesses.

---

> ### Author Response · Authors · 2024-11-25
> **Responses for Reviewer i3qN's Comments**
>
> **W1:** *The paper shows that the proposed model’s performance decreases dramatically with increasing noise level, with the PPI dataset. Although the proposed method performs better than other methods generally. The performance decrease rate is similar or even larger than other methods, as shown in Fig 2 (left).*
>
> **Response:** Thank you for the comment. We acknowledge that the performance of our proposed model decreases with increasing noise levels on the PPI dataset. This outcome is anticipated, as higher noise levels inherently obscure meaningful patterns. We also recognize this as a limitation of our approach.
>
> Despite this, as shown in Table 4 of the Appendix, our OPGM model performs better than its variant OPGM-rs, with a comparatively lower rate of performance degradation under increasing noise.
>
> **W2:** *It seems the approach’s performance is sensitive to the unbalancedness parameter ($\rho$), as shown in sensitivity analyses.*
>
> **Response:** Thank you for pointing this out. $\rho$ is indeed a critical component in unbalanced optimal transport literature [1], including optimal partial transport [2], as it governs the level of partiality allowed in the matching process. Consequently, it is expected that the model's performance is sensitive to $\rho$ given its pivotal role in balancing the degree of partiality of the final matching between the two graphs.
>
> To address this sensitivity, we aim to develop heuristics for setting $\rho$ based on the properties of input graphs, such as the partiality ratio between graphs in the training set.
>
> [1] T. S´ejourn´e, G. Peyr´e, and F.-X. Vialard, “Unbalanced optimal transport, from
> theory to numerics,” Handbook of Numerical Analysis, vol. 24, pp. 407–471, 2023.
>
> [2] Y. Bai, B. Schmitzer, M. Thorpe, and S. Kolouri, “Sliced optimal partial trans-
> port,” in Proceedings of the IEEE/CVF Conference on Computer Vision and Pat-
> tern Recognition, 2023, pp. 13 681–13 690.
>
> **W3:** *For applications where annotations are inconsistent or sparse, such as biological or social network datasets, the model’s dependence on well-annotated data could limit its generalizability.*
>
>
> **Response:** Thank you for your comment.  Our method demonstrates strong performance on sparse datasets, such as the IMCPT sparse dataset, and biological datasets like the PPI network dataset, outperforming other graph matching baselines.
> However, we acknowledge that our approach slightly underperforms on benchmarks like Pascal VOC Keypoint, which are known to contain annotation errors. Since our method relies on end-to-end supervised learning, inconsistencies or errors in annotations can adversely impact both the training process and evaluation metrics, which are calculated based on the ground truth.
>
> As this work focuses on developing an end-to-end partial graph matching framework, addressing annotation inconsistencies was not a primary objective. Nonetheless, we recognize this as a limitation and an avenue for future research to enhance the model's robustness in scenarios with inconsistent or sparse annotations.

---

### Author Response · Authors · 2024-11-25
**Summary of Changes**

We sincerely thank the reviewers for the constructive feedback. In response, we have made the following revisions in the updated manuscript:

- We revised the title of the manuscript.
- We revised the introduction to provide a clearer explanation of our approach.
- We updated the related work section by adding a discussion on the limitations of existing approaches, including optimal transport techniques for graph matching.
- Upon further inspection of our implementation, we found an inconsistency in the loss term that was given on line 390. We updated this loss function both in the manuscript and the code, and reran the numerical experiments. We found only very marginal differences in performance; some tests slightly performed better, and some slightly worse.
- We changed the term "feasibility threshold" to "feasibility condition" in order to emphasize that the "feasibility condition" is a property of our problem formulation, and we do not use a manual thresholding scheme.

---

### Meta-Review · Area_Chair_Peh8 · 2024-12-22

**Metareview:**

This submission proposes a method for partial graph matching. The proposed approach is based on the formulated objective based on optimal partial transport. For this objective, the paper proposes a solver (with theoretic guarantees), which can be integrated into an end-to-end learning framework by learning the matching bias and cost matrix. The paper is well motivated with an interesting formulation and the presented evaluation shows that the proposed approach can improve the computational efficiency.
The main criticism is on the large number of hyperparameters that need to be adjusted, potentially making practical adaptations to new settings hard. The hyperparameter \rho seems to be particularly important and the authors suggest a heuristic way to set it - without any guarantees.
Overall, the paper is a borderline case, with an interesting theoretic formulation.

**Additional Comments On Reviewer Discussion:**

The paper has received an average score of 5.75 with all reviews being on borderline scores. After the revision, which involves an update of the paper title, three out of four reviews are towards accepting the paper. Some criticism and extended discussion focused on the novelty of the proposed formulation, in particular by reviewer aXCT. The authors provide an extensive analysis on this matter and delineate their formulation of partial transport from previous works. After this discussion, the paper is a borderline accept case.

---

### Decision · Program_Chairs · 2025-01-22

Accept (Poster)